# Dynamic Stratified Contrastive Learning with Upstream Augmentation for MILP Branching

**Tongkai Lu**[1]  **Shuai Ma**[1]  **Chongyang Tao**[1]

## Abstract

Mixed Integer Linear Programming (MILP) is a fundamental NP-hard problem that has garnered significant attention from both academia and industry. The Branch-and-Bound (B&B) algorithm is the dominant approach for solving MILPs, where branching decisions play a critical role and have recently been enhanced by neural methods. However, these methods still struggle with semantic variation across depths, the scarcity of upstream nodes, and the costly collection of strong branching samples. To address these issues, we propose *SC-MILP*, a Dynamic **S**tratified **C**ontrastive Training Framework for **MILP** Branching. Our method groups B&B nodes based on their feature distributions and learns depth-aware, fine-grained node representations through dynamic stratified contrastive training. To address data scarcity and imbalance at upstream nodes, we introduce an upstream-augmented MILP derivation procedure that generates both theoretically equivalent and perturbed instances. Experiments on both synthetic and real-world MILP benchmarks, including large-scale instances, show that *SC-MILP* significantly improves branching accuracy, reduces solving time, with particularly strong gains at upstream nodes.

## 1. Introduction

Mixed-Integer Linear Programming (MILP) is a fundamental class of NP-hard combinatorial optimization problems that integrates discrete and continuous decision variables under linear constraints (Bénichou et al., 1971). It not only plays an important role in solving other combinatorial optimization problems, but also serves as a powerful modeling framework for a wide range of real-world applications, in-cluding scheduling, network design, routing, and strategic planning (Zhao et al., 2025; Wang & Li, 2024; Guo et al., 2025; Floudas & Lin, 2005; Yan et al., 2024; Gao et al., 2024). The Branch-and-Bound (B&B) algorithm is the dominant exact method for solving MILPs, constructing a search tree by recursively partitioning the feasible space (Land & Doig, 1960). Each B&B node [1] represents a subproblem defined by additional branching constraints.

Variable selection (branching) is the most critical decision in B&B, as it directly determines which variable to branch on at each node and thereby influences the size of the search tree and the overall solving time (Fischetti & Lodi, 2003). Traditional variable selection methods rely heavily on expert-crafted rules and are computationally expensive, making them inflexible and time-consuming. More recently, motivated by the success of neural network methods in other combinatorial optimization domains, researchers have begun to explore their application to variable selection. Gasse et al. (2019) map MILPs to bipartite graphs and employ Graph Convolutional Neural Networks (GCNNs) to extract features of candidate variables for branching. Their approach adopts imitation learning, treating each B&B node as a training sample and targeting strong branching (Applegate et al., 1995), which is widely used to construct smaller B&B trees. This work demonstrated notable performance gains and established a line of research on neural branching strategies (Alvarez et al., 2017; Burges, 2010; Seyfi et al., 2023; Gupta et al., 2020; Zarpellon et al., 2021; Khalil et al., 2022; Lin et al., 2022; Li et al., 2025; Huang et al., 2024; Lin et al., 2024; Wang et al., 2024).

Despite this progress, learning effective branching policies remains challenging, especially at upstream nodes near the root of the B&B tree. First, nodes at different depths correspond to distinct solving stages, leading to systematic semantic shifts and distributional changes across the tree (Appendix A.1). This continuous variation ultimately produces large disparities between upstream and downstream nodes. However, existing neural-based branching methods largely treat all nodes uniformly, obscuring depth-dependent

---

[1]SKLCCSE, Beihang University, Beijing 100191, China. Correspondence to: Shuai Ma <shuaima@buaa.edu.cn>.

*Proceedings of the 43rd International Conference on Machine Learning*, Seoul, South Korea. PMLR 306, 2026. Copyright 2026 by the author(s).

[1]In this paper, "node" refers specifically to a B&B tree node (a MILP subproblem/training instance), while "vertex" refers to a node in the MILP's bipartite graph.

variations and degrading branching accuracy. Second, the inherent imbalance of the B&B tree results in far fewer upstream nodes than downstream ones. Consequently, training is dominated by downstream nodes, impairing performance on the critical upstream nodes that largely determine solving efficiency. Third, acquiring strong branching supervision is computationally expensive (Appendix A.2), further exacerbating the scarcity of high-quality upstream data.

To address these challenges, we propose *SC-MILP*, a Dynamic **S**tratified **C**ontrastive Training Framework for **MILP** Branching. Our method organizes B&B nodes into stratified groups based on their feature distributions and applies a contrastive objective that progressively separates node representations across groups, enabling the model to capture fine-grained, depth-dependent semantics variations throughout the B&B tree. To mitigate data scarcity and imbalance at upstream nodes, we further introduce an upstream-augmented MILP derivation procedure that systematically generates both theoretically equivalent and perturbed MILPs from the original instances, enriching early-stage training data without additional strong branching costs. By integrating these strategies, SC-MILP effectively captures subtle semantic differences between nodes, thereby enhancing branching accuracy and the solving efficiency. The main contributions of this work are as follows:

(1) We propose *SC-MILP*, a novel training framework that first groups nodes according to their feature distributions and then trains a GCNN-based discriminative model to effectively capture semantic variations within B&B trees.

(2) We design an *upstream-augmented MILP derivation* procedure that systematically generates both theoretically equivalent MILPs and perturbed variants from the original instances to address data scarcity and imbalance at upstream nodes without the extra cost of collecting strong branching expert samples.

(3) We present a *dynamic stratified contrastive learning* that contrasts nodes within and across groups, with separation progressively increasing along the group hierarchy. This enables the learning of finer-grained node representations, leading to more informed and effective branching decisions.

(4) Our *SC-MILP* significantly improves MILP solving efficiency, outperforming all traditional branching strategies and neural-based methods. Moreover, it enhances branching node prediction accuracy, with particularly pronounced gains at upstream nodes.

## 2. Related Work

In this section, we concentrate on neural-based methods for MILP branching, as this represents the primary focus of our study; other neural approaches for solving MILP problems are discussed in Appendix B.

### 2.1. Neural-based Methods for MILP Branching

**Supervised learning based methods.** Strong branching (Applegate et al., 1995) yields highly effective decisions but is computationally expensive, motivating neural-based approximations. Early approaches train machine learning models to imitate strong branching using expert labels (Burges, 2010; Alvarez et al., 2017; Marcos Alvarez et al., 2014; Khalil et al., 2016), with GCNN-based bipartite graph representations becoming a dominant paradigm (Khalil et al., 2016). Subsequent works improve efficiency or accuracy by enhancing model architectures or training strategies, including hybrid GNN–MLP designs (Gupta et al., 2020), temporal attention mechanisms (Seyfi et al., 2023), higher-order GNNs (Chen et al., 2024), and data-centric generalization via LLM-generated MILPs (Li et al., 2025). To alleviate label scarcity, CAMBranch (Lin et al., 2024) introduces variable-shifting augmentation; however, its reliance on a single augmentation strategy limits data diversity and increases overfitting risk. Beyond strong branching imitation, several works focus on learning pseudo-cost branching (Zarpellon et al., 2021; Pei & Chen, 2023; Khalil et al., 2022; Lin et al., 2022), which rely on indirect heuristic signals and are less effective.

**Reinforcement Learning based Methods.** Reinforcement learning–based methods model branching as a finite-horizon Markov Decision Process (MDP). Early approaches apply value or policy-based RL to directly learn branching decisions (Etheve et al., 2020; Scavuzzo et al., 2022; Zhang et al., 2022). Subsequent works improve learning efficiency and exploration by exploiting search trajectories, such as Retro (Parsonson et al., 2023) and SORREL (Feng & Yang, 2025). On the other hand, Symb4CO (Kuang et al., 2024a), GS4CO (Kuang et al., 2024b) and DQN-BBMDP (Strang et al., 2025) use reinforcement learning to devise branching algorithms and combine symbolic learning to enhance robustness. However, these methods remain constrained by sparse rewards and credit assignment.

### 2.2. Contrastive Learning in MILP/ILP Solving

Some studies have also explored Contrastive learning (CL) for MILP solving. Existing contrastive-learning-based methods for MILP and ILP solving largely adopt an outcome-driven paradigm, where positive samples correspond to desirable solutions or solver configurations and negatives to inferior ones, as in ConPaS (Huang et al., 2024), CL-LNS (Huang et al., 2023), and CLCR (Zeng et al., 2025). While effective for modeling solution quality or configuration preferences, these approaches do not explicitly target branching decisions. CAMBranch (Lin et al., 2024) applies contrastive learning to branching by pulling together

representations of original instances and their augmentations, but its narrow definition of positive pairs overlooks intrinsic similarities across different samples and may separate semantically similar representations. *In contrast, our method structures contrastive learning through dynamic stratification, treating samples within the same stratum as positives and across strata as negatives, thereby capturing both similarity and heterogeneity in branching semantics and enabling more robust strong branching prediction.*

## 3. Preliminaries

**Mixed Integer Linear Programming (MILP).** We consider the standard form of a Mixed Integer Linear Program (MILP), defined as follows:

$$\min_{\mathbf{x}} \ \mathbf{c}^{\top}\mathbf{x} \quad \text{s.t.} A\mathbf{x} \leq \mathbf{b}, \ \mathbf{l} \leq \mathbf{x} \leq \mathbf{u}, \ x_j \in \mathbb{Z}, \ \forall j \in \mathcal{I}, \quad (1)$$

where $\mathbf{x} = \{x_1, ..., x_n\} \in \mathbb{R}^n$ is the decision variable vector, $\mathbf{c} \in \mathbb{R}^n$ is the objective coefficient vector, $A \in \mathbb{R}^{q \times n}$, $\mathbf{b} \in \mathbb{R}^q$ define the system of linear inequality constraints, $\mathbf{l} \in (\mathbb{R} \cup \{-\infty\})^n$, $\mathbf{u} \in (\mathbb{R} \cup \{+\infty\})^n$ are the lower and upper bounds and $\mathcal{I} \subseteq \{1, 2, \ldots, n\}$ is the index set of variables constrained to be integers.

**Branching for MILPs.** In a Branch-and-Bound (B&B) tree $\mathcal{T}$, each node $s$ represents a subproblem with candidate set $\text{Cand}(s)$. The goal is to learn a scoring function

$$\psi(s, x) : \mathcal{S} \times \text{Cand}(s) \to \mathbb{R}, \quad (2)$$

where $\mathcal{S}$ is the set of all nodes in $\mathcal{T}$. The optimal branching candidate is then

$$x^*(s) = \arg \max_{x \in \text{Cand}(s)} \psi(s, x). \quad (3)$$

**Bipartite Graph Representation.** Following standard MILP branching literature (Gasse et al., 2019; Gupta et al., 2020; Zarpellon et al., 2021; Lin et al., 2022; Seyfi et al., 2023; Lin et al., 2024; Li et al., 2025), we treat each B&B node as a training instance. Each MILP instance is modeled as a bipartite graph $(\mathcal{G}, C, E, V)$, where an edge $(i, j) \in E$ exists if variable $x_j$ appears in constraint $i$. Constraint and variable features are stored in $C \in \mathbb{R}^{|C| \times d_1}$ and $V \in \mathbb{R}^{|V| \times d_2}$, and edge features in $E \in \mathbb{R}^{|C| \times |V| \times d_3}$. Details of these features are in Appendix D.2.1.

## 4. Method

### 4.1. Overview

In this section, we present *SC-MILP*, a dynamic stratified contrastive training framework for accurate branching and efficient MILP solving (Fig. 1). We first perform stratified node grouping to partition B&B nodes into feature-driven strata, capturing continuous structural and stage-wise variations. To mitigate upstream scarcity, we introduce upstream-augmented MILP derivation, which generates both theoret-

ically equivalent MILPs (via linear transformations) and perturbed MILPs (via lightweight perturbations to objectives, constraints, and dual variables), enriching the training distribution without additional data collection. Finally, we apply dynamic stratified contrastive learning to leverage semantic and feature variations across the tree, where nodes within the same stratum are treated as positives and those from other strata as negatives, with similarity scores dynamically reweighted by stratum depth differences. This design preserves intra-stratum consistency, encourages gradual separation between adjacent strata, and enforces stronger discrimination across distant strata, leading to more accurate and robust branching decisions.

### 4.2. Stratified Node Grouping

Considering that nodes at similar depths share comparable solving contexts, we perform *stratified node grouping* that partitions B&B nodes into $m$ strata $\{G_1, \ldots, G_m\}$ based on their feature distributions, capturing structural and semantic variations across the search tree. The groups are obtained via unsupervised clustering (e.g., K-means[2]), with $m$ selected using the elbow method. This principled grouping establishes a clear stratification that underpins subsequent upstream-augmented MILP derivation and dynamic stratified contrastive learning. Further implementation details are provided in Appendix E.4.1.

### 4.3. Upstream-Augmented MILP Derivation

Due to the structural imbalance of the B&B tree, upstream nodes are substantially fewer than downstream ones, which limits the model's ability to learn discriminative semantics at early stages and degrades dynamic stratified contrastive learning. After stratified grouping, this imbalance is further amplified across strata. While regenerating strong branching expert samples could mitigate this issue, it is computationally expensive and may distort the feature distribution of the original groups, impairing model learning.

To address these issues, we propose *upstream-augmented MILP derivation*, which systematically generates new training samples from the original MILPs. Specifically, the Equivalence-based derivation applies linear transformations to produce formally distinct but feasible-equivalent samples with identical strong branching labels, while perturbation-based derivation slightly modifies constraints and dual variables to improve generalization and robustness. Unlike standard data augmentation, our approach is grounded in MILP structure, with a theoretical correspondence between augmented and original bipartite graphs (Appendix D.2), ensuring fidelity and soundness.

---

[2]We adopt K-means as it provides a simple yet effective solution for our task; alternative clustering methods did not yield additional benefits in our experiments in Appendix D.1.

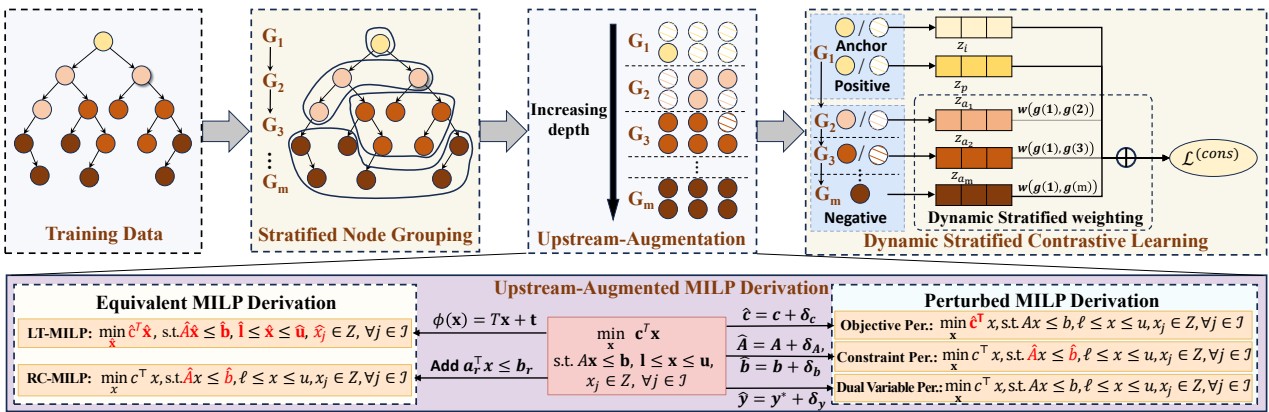

*Figure 1.* Overview of *SC-MILP*. Nodes in the B&B tree are color-coded to reflect feature variations across depths, while slashed nodes denote additional samples generated via upstream-augmented MILP derivation. Nodes are first partitioned into feature-driven strata $(G_1, G_2, \dots)$. Both equivalent and perturbed derivations are then applied to augment upstream samples: *LT-MILP* and *RC-MILP* denote linear-transformation– and redundant-constraint–based derivations, while objective Per., constraint Per., and dual variable Per. indicate lightweight variations to the objective, constraints, and dual variables, respectively. Finally, dynamic stratified contrastive learning is applied, treating nodes within the same stratum as positives and those from different strata as negatives, with stratified weighting that increases separation with stratum depth and is adaptively modulated during training.

### 4.3.1. EQUIVALENT MILP DERIVATION

We exploit expert knowledge in original MILPs and introduce two equivalent derivation methods: linear transformation (Definition 4.1) and redundant constraint–based derivation (Definition 4.4). Both are supported by theoretical guarantees (Theorems 4.2 & 4.3 & C.1, proofs in Appendix C.2). These establish a one-to-one correspondence between the feasible regions (including integer variables) and strong branching outcomes of the original and derived MILPs, thus preserving fidelity and theoretical rigor.

**Definition 4.1.** [Linear transformation based Equivalent MILP Derivation (*LT-MILP*)] Given an MILP problem in Equ. 1, let $\phi(\mathbf{x}) = T\mathbf{x} + \mathbf{t}$ be an affine transformation, where $T$ is a diagonal matrix with $T_{jj} \in \{1, -1\}$ and $\mathbf{t}_{\mathcal{I}} \in \mathbb{Z}^k$. This induces an equivalent MILP:

$$\min_{\hat{\mathbf{x}}} \ \hat{\mathbf{c}}^{\mathsf{T}}\hat{\mathbf{x}} \text{ s.t. } \hat{A}\hat{\mathbf{x}} \leq \hat{\mathbf{b}}, \ \hat{\mathbf{l}} \leq \hat{\mathbf{x}} \leq \hat{\mathbf{u}}, \ \hat{x}_j \in \mathbb{Z}, \ \forall j \in \mathcal{I}, \quad (4)$$

where $\hat{\mathbf{c}} = T\mathbf{c}$, $\hat{A} = AT$, $\hat{\mathbf{b}} = \mathbf{b} + AT\mathbf{t}$, and the transformed bounds are defined by $\hat{l}_j = \min\big((T\mathbf{l} + \mathbf{t})_j, (T\mathbf{u} + \mathbf{t})_j\big)$, $\hat{u}_j = \max\big((T\mathbf{l} + \mathbf{t})_j, (T\mathbf{u} + \mathbf{t})_j\big)$, $\forall j$.

Next, we present two theorems establishing the equivalence between *LT-MILP* and the original MILP problem, with proofs given in Appendix C.2.

**Theorem 4.2.** *Let $\mathcal{P} := \{\mathbf{x} \in \mathbb{R}^n \mid A\mathbf{x} \leq \mathbf{b}, \ \mathbf{l} \leq \mathbf{x} \leq \mathbf{u}\}$ be the feasible region of the LP relaxation of Eq. 1, and let $\hat{\mathcal{P}} := \{\hat{\mathbf{x}} \in \mathbb{R}^n \mid \hat{A}\hat{\mathbf{x}} \leq \hat{\mathbf{b}}, \ \hat{\mathbf{l}} \leq \hat{\mathbf{x}} \leq \hat{\mathbf{u}}\}$ be the feasible region of the LP relaxation of Eq. 4. Then: (1) The affine transformation $\phi(\cdot)$ is a bijection between $\mathcal{P}$ and $\hat{\mathcal{P}}$, i.e., the LP feasible domains are in one-to-one correspondence. (2) This bijection $\phi(\cdot)$ induces a one-to-one correspondence between the sets of optimal solutions of the two LPs.*

**Theorem 4.3.** *For any variable $x_j$ in the original problem of Equ. 1, the strong branching decisions correspond exactly to those of $\hat{x}_{\pi(j)}$ in the transformed problem of Equ. 4, where $\pi$ is the variable mapping induced by $\phi(\cdot)$.*

Theorems 4.2 and 4.3 establish equivalence between the original and transformed problems in both feasible regions and strong branching outcomes. To further broaden the theoretical scope and enable more diverse augmented MILPs, we generalize the linear transformation assumption to a broader class of affine mappings (Appendix C.1).

**Definition 4.4.** [Redundant Constraint based Equivalent MILP Derivation (*RC-MILP*)] Given an MILP problem in Eq. 1, let $a_i^{\mathsf{T}} x \leq b_i$ and $a_j^{\mathsf{T}} x \leq b_j$ be two linearly independent constraints; define a new redundant constraint $a_r^{\mathsf{T}} x \leq b_r$ with $a_r = a_i + a_j$, $b_r = b_i + b_j$. Augmenting the MILP with this constraint yields $\min_x \mathbf{c}^{\mathsf{T}} x$ s.t. $\hat{A}\mathbf{x} \leq \hat{b}, \mathbf{l} \leq \mathbf{x} \leq \mathbf{u}, x_j \in \mathbb{Z}, \forall j \in \mathcal{I}$, where $\hat{A} = [A, a_r^{\mathsf{T}}]^T, \hat{b} = [b, b_r]^T$.

As the constraints of *RC-MILP* are essentially equivalent to those of the original problem, their LP relaxations share exactly the same feasible region. Consequently, they produce identical strong branching results and optimal solutions. Therefore, *RC-MILP* serves as an equivalent derivation of the original MILP.

### 4.3.2. PERTURBED MILP DERIVATION

To further diversify problem structures, we introduce a perturbation-based derivation that generates non-identical MILPs via small perturbations to objective coefficients, constraint coefficients, or dual variables. Although such perturbations may slightly affect branching behavior, we retain the

original strong branching labels to encourage more robust representations, shown in Definitions 4.5 & 4.6.

**Definition 4.5.** [Objective and Constraint Perturbation] Given an MILP in standard form, we consider two types of perturbations to generate derived MILPs:

**(1) Objective Perturbation:** The objective vector is perturbed as $\hat{\mathbf{c}} = \mathbf{c} + \delta_c$, where $\delta_c \in \mathbb{R}^n$ is the Gaussian noise. The augmented MILP becomes

$$\min_{\mathbf{x}} \ \hat{\mathbf{c}}^\mathsf{T}\mathbf{x} \quad \text{s.t.} \quad A\mathbf{x} \le \mathbf{b}, \ \mathbf{l} \le \mathbf{x} \le \mathbf{u}, \ x_j \in \mathbb{Z}, \ \forall j \in \mathcal{I}$$

**(2) Constraint Perturbation:** The constraint system is perturbed as $\hat{A} = A + \delta_A, \quad \hat{\mathbf{b}} = \mathbf{b} + \delta_b$, where $\delta_A \in \mathbb{R}^{q \times n}$ and $\delta_b \in \mathbb{R}^q$ are Gaussian noises. The augmented MILP is

$$\min_{\mathbf{x}} \ \mathbf{c}^\mathsf{T}\mathbf{x} \quad \text{s.t.} \quad \hat{A}\mathbf{x} \le \hat{\mathbf{b}}, \ \mathbf{l} \le \mathbf{x} \le \mathbf{u}, \ x_j \in \mathbb{Z}, \ \forall j \in \mathcal{I}$$

**Definition 4.6.** [Dual Variable Perturbation] For the LP relaxation of the MILP, let $y^* \in \mathbb{R}^q$ denote the optimal dual variables corresponding to constraints $Ax \le b$. We perturb the dual variables by injecting Gaussian noise $\delta_y$

$$\hat{y} = y^* + \delta_y,$$

### 4.4. Dynamic Stratified Contrastive Learning

In a Branch-and-Bound (B&B) tree, node feature distributions vary substantially across depths, reflecting different solving stages. For instance, `sol_is_at_lb` and `sol_is_at_ub`—indicating whether a variable's LP solution matches its lower (upper) bound—are typically inactive at upstream nodes but become dominant downstream as variables are progressively fixed. This is because upstream nodes capture global optimization potential, whereas downstream nodes are strongly shaped by prior branching decisions. Learning a single shared representation across all depths therefore blurs critical upstream semantics and degrades branching accuracy and generalization.

A straightforward alternative is supervised contrastive learning (SCL) (Khosla et al., 2020), which pulls samples from the same group together and treats all others as negatives. However, SCL ignores the gradual semantic transitions across depths: nodes from adjacent stages, though semantically closer, are pushed apart as strongly as distant ones. This abrupt separation conflicts with the progressive, stratified nature of B&B search and disproportionately harms the learning of underrepresented upstream nodes.

To address this issue, we propose a *dynamic stratified contrastive learning* framework that explicitly models depth-aware semantic variation. Unlike conventional SCL, our approach assigns dynamic stratified weights to contrastive pairs based on group distance, keeping nearby nodes moderately close while enforcing stronger separation between distant ones. This strategy preserves intra-group consistency, encourages smooth transitions across adjacent strata, and

enhances representation quality for branching decisions.

**Dynamic Stratified Contrastive Loss.** Given an MILP node $s_i = (\mathcal{G}_i, C_i, E_i, V_i)$, we first encode its bipartite graph (node) using a GCNN to obtain variable embeddings, and then aggregate these embeddings via mean pooling to form a graph-level representation $Z_i = \text{MeanPool}\{z_v \mid v \in V_i\}$. The dynamic stratified contrastive loss is defined as

$$\mathcal{L}^{(cons)} = \sum_i -\frac{1}{|P(i)|} \sum_{p \in P(i)} \log \frac{\exp\left(f\left(Z_i, Z_p\right)/\tau\right)}{sum_i + \sum_{j \in P(i)} \exp(f(Z_i, Z_j)/\tau)}$$

where $sum_i = \sum_{j \in N(i)} \exp(w(g(i), g(j)) \cdot f(Z_i, Z_j)/\tau)$, $P(i) = \{p \ne i \mid g(p) = g(i)\}$ are positive nodes from the same group, $N(i) = \{p \ne i \mid g(p) \ne g(i)\}$ are negative nodes, $f(\cdot, \cdot)$ denotes cosine similarity, $\tau$ is the temperature.

The key component of our formulation is the dynamic stratified weight $w(g(i), g(j))$, which explicitly controls the repulsion strength between nodes from different strata:

$$w(g(i), g(j)) = \sigma\Big( \sum_{l=1}^{|g(i) - g(j)|} \log(1 + \exp(\theta_l)) \Big), \quad (5)$$

where $\theta_l$ are learnable parameters, and the sigmod function $\sigma$ bounds the weight in $(0, 1)$. This formulation is differentiable with respect to $\theta_l$, and the cumulative softplus structure induces a monotonic increase of $w(g(i), g(j))$ with the relative stratum distance $|g(i) - g(j)|$.

In supervised contrastive learning, an anchor embedding $\mathbf{Z}_i$ is encouraged to move closer to all positive samples while being pushed away from negative samples by maximizing the ratio of positive similarity to total similarity. By assigning a distance-aware weight $w > 0$ to each negative sample, its contribution to the denominator is explicitly modulated, so that larger weights induce stronger repulsion and larger embedding separation. Note that, even if $w < 1$, the negative sample is still repelled, as its contribution remains positively weighted in the loss. As a result, this weighting scheme allows us to smoothly control the degree of inter-stratum separation, while preserving the relative closeness of samples from the same stratum.

**Supervised Branching Loss.** Following the imitation learning paradigm of Gasse et al. (2019), we train the branching policy to imitate Strong Branching decisions. Expert trajectories are collected using the SCIP optimization suite (Gamrath et al., 2020), resulting in a dataset of expert state–action pairs $\mathcal{D} = \{(s_i, a_i^*)\}_{i=1}^N$. Given an MILP state $s_i$, $a_i^*$ denotes its branching decision given by the Strong Branching strategy. The policy network is optimized using the cross-entropy loss

$$\mathcal{L}^{(\text{sup})} = -\frac{1}{N} \sum_{(s_i, a_i^*) \in \mathcal{D}} \log \pi_\theta(a_i^* \mid s_i). \quad (6)$$

**Overall Training Loss.** The overall training objective com-

bines the original supervised loss $\mathcal{L}^{(sup)}$ with the dynamic stratified contrastive loss:

$$\mathcal{L} = \mathcal{L}^{(sup)} + \lambda\mathcal{L}^{(cons)}, \qquad (7)$$

where $\lambda$ is a learnable parameter as a balancing coefficient, set to 0 during inference.

### 4.5. Branching with *SC-MILP*

During MILP solving, *SC-MILP* is invoked at every B&B branching decision. However, at upstream nodes near the root, where branching decisions largely determine search time and tree size, we adopt a hybrid strategy: *SC-MILP* first scores all candidate variables to select the top-$k$, and strong branching chooses the best among them. At downstream nodes, *SC-MILP* is used directly for branching. Since the total tree depth is unknown, we distinguish upstream and downstream nodes using the size of the candidate set, which reflects both solving stage and branching difficulty. Specifically, we define the cutoff depth $d_0$ as the shallowest depth at which the number of candidates falls below a fraction $\rho$ of the root candidates, *i.e.*, $d_0 = \min\{d \mid n_{\text{cand}}(d) \leq \rho \cdot n_{\text{root}}\}$, where $n_{\text{cand}}(d)$ denotes the number of branching candidates at depth $d$ and $\rho \in (0, 1)$ is a hyperparameter.

## 5. Experiments

We conduct an extensive experimental study to validate the advantage of our *SC-MILP*. Full experimental details are provided in Appendix E, and training and inference efficiency analysis is reported in Appendix F.

### 5.1. Experimental Settings

**Datasets.** Following Gasse et al. (2019); Seyfi et al. (2023); Lin et al. (2024); Li et al. (2025), we evaluate *SC-MILP* on four standard NP-hard MILP benchmarks—Set Covering (Balas & Ho, 2009), Combinatorial Auction (Leyton-Brown et al., 2000), Capacitated Facility Location (Cornuéjols et al., 1991), and Maximum Independent Set (Bergman et al., 2016). Instances are generated from SCIP rollouts with Strong Branching at three difficulty levels (Easy/Medium/Hard); models are trained on 100K easy-level expert samples augmented with ~30% upstream-derived data, and evaluated on 100 instances per level.

We also evaluate *SC-MILP* on a diverse collection of real-world MILP benchmarks to assess its generalization ability and practical performance. The evaluated datasets include MIK (Atamtürk, 2003), which consists of mixed-integer knapsack instances; CORLAT (Gomes et al., 2008), containing network design problems with complex combinatorial structures; the MIPLIB mixed neos and mixed supportcase instances derived from MIPLIB (Gleixner et al., 2021), representing a wide range of real-world MILPs with

highly heterogeneous characteristics[3]; and the Load Balancing and Anonymous problem classes from the NeurIPS 2021 ML4CO competition (Bengio et al., 2021), which reflect practical operational optimization scenarios.

For all datasets, we randomly split the instances into training and test sets with an 80% / 20% ratio. Detailed statistics of the datasets are summarized in Table 9.

**Baselines.** Following common practice, we restrict evaluation to nueral-based branching methods, as other MILP strategies differ in solving stages, evaluation criteria, or solver configuration, including: (1) conventional methods full strong branching (FSB) (Applegate et al., 1995) and Reliability Pseudocost Branching (RPB) (Achterberg et al., 2005), (2) machine learning based methods Trees (Alvarez et al., 2017), LMART (Burges, 2010) and svmrank (Khalil et al., 2016), (3) neural based methods GCNN (Gasse et al., 2019), FILM (Gupta et al., 2020), TreeGate (Zarpellon et al., 2021), T-BranT (Lin et al., 2022), TGAT (Seyfi et al., 2023), CAMBranch (Lin et al., 2024), MILP-Evolve (Li et al., 2025), and (4) RL based methods Symb4CO (Kuang et al., 2024a) , GS4CO (Kuang et al., 2024b) and DQN-BBMDP (Strang et al., 2025).

**Metrics**. Following standard MILP benchmarks, we report solving time, the number of explored B&B nodes, and the number of instances where each method achieves the best solving time (wins). Lower time and node counts are preferred, with solving time being the primary metric. We further report top-$k$ accuracy (acc@1/3/5/10) against strong branching to assess branching decision quality.

**Implementation details.** Our experiments were conducted on an Intel Xeon Gold 6148 CPU@2.40GHz and an NVIDIA Tesla V100 PCIe 32GB GPU. The number of strata is set to $m = 4$ for Set Covering, Combinatorial Auction, and Maximum Independent Set, and $m = 5$ for Capacitated Facility Location, determined via the elbow method. Upstream augmentation applies two equivalent MILP derivations with 35% probability and three perturbation-based derivations with 10% probability. Based on validation results, we set the contrastive temperature $\tau = 0.08$, the hybrid branching ratio $\rho = 0.8$, and $k = 5$. Training follows standard configurations from prior work (Khosla et al., 2020; Gasse et al., 2019; Gupta et al., 2020; Zarpellon et al., 2021; Lin et al., 2024), and all baselines use their released implementations with default parameters.

All experiments are conducted using SCIP 7.0.1 (Gamrath et al., 2020). Following standard evaluation protocols, in-

---

[3]Following prior work (Turner et al., 2023), directly learning on the full MIPLIB benchmark is extremely challenging due to the high heterogeneity of instances, which limits the effectiveness of learning-based methods. We therefore adopt two representative subsets constructed by Wang et al. (2023).

*Table 1.* Performance on generated datasets. The best-performing methods in terms of Wins and Time are highlighted in bold, while for Nodes, the best neural-based method is highlighted.

| | Set Covering | | | | | | | | | Combinatorial Auction | | | | | | | | |
| --- | --- | --- | --- | --- | --- | --- | --- | --- | --- | --- | --- | --- | --- | --- | --- | --- | --- | --- |
| | Easy | | | Medium | | | Hard | | | Easy | | | Medium | | | Hard | | |
| Model | Wins↑ | Time↓ | Nodes↓ | Wins↑ | Time↓ | Nodes↓ | Wins↑ | Time↓ | Nodes↓ | Wins↑ | Time↓ | Nodes↓ | Wins↑ | Time↓ | Nodes↓ | Wins↑ | Time↓ | Nodes↓ |
| FSB (Applegate et al., 1995) | 0/100 | 17.42 | 16 | 0/100 | 409.32 | 164 | 0/100 | 3600 | n/a | 0/100 | 4.12 | 6 | 0/100 | 87.45 | 72 | 0/100 | 1821.62 | 401 |
| RPB (Achterberg et al., 2005) | 0/100 | 8.91 | 55 | 0/100 | 59.73 | 1734 | 0/100 | 1654.84 | 47352 | 0/100 | 2.73 | 10 | 0/100 | 21.87 | 687 | 0/100 | 137.11 | 5472 |
| Trees (Alvarez et al., 2017) | 0/100 | 9.31 | 187 | 0/100 | 92.66 | 4203 | 0/100 | 1800.62 | 45126 | 0/100 | 2.51 | 87 | 0/100 | 23.66 | 980 | 0/100 | 458.35 | 10183 |
| LMART (Burges, 2010) | 0/100 | 7.21 | 171 | 0/100 | 59.86 | 1903 | 0/100 | 1252.01 | 34331 | 0/100 | 1.98 | 75 | 11/100 | 17.42 | 876 | 0/100 | 224.02 | 6149 |
| svmrank (Khalil et al., 2016) | 0/100 | 8.07 | 163 | 0/100 | 73.06 | 1937 | 4/100 | 1038.14 | 31089 | 0/100 | 2.34 | 78 | 0/100 | 23.16 | 868 | 0/100 | 376.61 | 6816 |
| GCNN (Gasse et al., 2019) | 11/100 | 6.11 | 171 | 4/100 | 42.44 | 1484 | 0/100 | 1299.99 | 37108 | 0/100 | 1.88 | 72 | 0/100 | 19.31 | 655 | 1/100 | 114.32 | 5231 |
| FILM (Gupta et al., 2020) | 3/100 | 6.29 | 165 | 0/100 | 44.32 | 1391 | 0/100 | 1392.42 | 33692 | 11/100 | 1.77 | 73 | 0/100 | 26.04 | 857 | 0/100 | 416.53 | 5310 |
| TreeGate (Zarpellon et al., 2021) | 0/100 | 8.32 | 231 | 0/100 | 51.4 | 2410 | 0/100 | 2085.85 | 58536 | 0/100 | 2.35 | 83 | 0/100 | 18.32 | 862 | 0/100 | 176.4 | 5437 |
| T-BranT (Lin et al., 2022) | 0/100 | 6.91 | 153 | 0/100 | 43.53 | 1653 | 0/100 | 1154.46 | 34694 | 0/100 | 2.28 | 89 | 0/100 | 19.56 | 723 | 0/100 | 142.73 | 6742 |
| TGAT (Seyfi et al., 2023) | 3/100 | 6.8 | 127 | 1/100 | 46.81 | 1336 | 1/100 | 1174.38 | 29452 | 0/100 | 2.01 | 75 | 3/100 | 22.03 | 694 | 0/100 | 126.49 | 5531 |
| Symb4CO (Kuang et al., 2024a) | 19/100 | 6.09 | 151 | 4/100 | 43.37 | 1438 | 0/100 | 1372.47 | 57315 | 29/100 | 1.69 | 75 | 5/100 | 17.34 | 743 | 2/100 | 108.7 | 5637 |
| GS4CO (Kuang et al., 2024b) | 2/100 | 6.21 | 216 | 1/100 | 42.74 | 1735 | 1/100 | 1147.84 | 63142 | 0/100 | 1.73 | 82 | 1/100 | 18.48 | 746 | 2/100 | 104.39 | 6482 |
| DQN-BBMDP (Strang et al., 2025) | 0/100 | 6.24 | 219 | 2/100 | 42.49 | 1632 | 1/100 | 1139.04 | 62794 | 0/100 | 1.75 | 84 | 0/100 | 18.17 | 736 | 2/100 | 105.17 | 6573 |
| CAMBranch (Lin et al., 2024) | 2/100 | 6.33 | 139 | 11/100 | 41.53 | **1279** | 4/100 | 1104.34 | 31584 | 4/100 | 1.77 | 87 | 3/100 | 17.79 | 683 | 1/100 | 125.94 | 4904 |
| MILP-Evolve (Li et al., 2025) | 0/100 | 10.31 | 144 | 3/100 | 46.37 | 1431 | 7/100 | 1024.21 | 30812 | 3/100 | 1.78 | **64** | 12/100 | 17.5 | 663 | 5/100 | 106.73 | 5316 |
| *SC-MILP* (Ours) | **60/100** | **5.99** | 117 | **74/100** | **37.01** | 1452 | **82/100** | **953.24** | 29375 | **53/100** | **1.63** | 68 | **65/100** | **16.85** | **645** | **87/100** | **99.81** | **4721** |
| | Capacitated Facility Location | | | | | | | | | Maximum Independent Set | | | | | | | | |
| | Easy | | | Medium | | | Hard | | | Easy | | | Medium | | | Hard | | |
| Model | Wins↑ | Time↓ | Nodes↓ | Wins↑ | Time↓ | Nodes↓ | Wins↑ | Time↓ | Nodes↓ | Wins↑ | Time↓ | Nodes↓ | Wins↑ | Time↓ | Nodes↓ | Wins↑ | Time↓ | Nodes↓ |
| FSB (Applegate et al., 1995) | 0/100 | 30.49 | 14 | 0/100 | 224.13 | 81 | 0/100 | 748.33 | 52 | 0/100 | 23.57 | 7 | 0/100 | 1581.86 | 38 | 0/100 | 3600 | n/a |
| RPB (Achterberg et al., 2005) | 0/100 | 26.37 | 23 | 0/100 | 157.73 | 169 | 0/100 | 645.72 | 105 | 0/100 | 11.33 | 21 | 0/100 | 111.41 | 731 | 0/100 | 2124.75 | 7815 |
| Trees (Alvarez et al., 2017) | 0/100 | 28.91 | 133 | 0/100 | 159.88 | 404 | 0/100 | 634.12 | 400 | 0/100 | 10.68 | 73 | 0/100 | 1178.31 | 4643 | 0/100 | 3442.23 | 28210 |
| LMART (Burges, 2010) | 0/100 | 23.36 | 114 | 0/100 | 129.1 | 357 | 0/100 | 520.26 | 345 | 14/100 | 8.31 | 52 | 0/100 | 219.22 | 747 | 0/100 | 3356.55 | 33732 |
| svmrank (Khalil et al., 2016) | 0/100 | 23.61 | 116 | 0/100 | 130.88 | 351 | 0/100 | 512.98 | 331 | 0/100 | 13.77 | 45 | 0/100 | 241.83 | **541** | 0/100 | 3174.23 | 20030 |
| GCNN (Gasse et al., 2019) | 0/100 | 22.15 | 107 | 0/100 | 121.31 | 341 | 0/100 | 563.54 | 345 | 0/100 | 11.44 | 43 | 0/100 | 192.86 | 1837 | 0/100 | 1187.5 | 18668 |
| FILM (Gupta et al., 2020) | 0/100 | 21.56 | 104 | 1/100 | 116.81 | 337 | 0/100 | 543.14 | 358 | 0/100 | 10.73 | 47 | 0/100 | 164.57 | 1682 | 0/100 | 3528.71 | 16667 |
| TreeGate (Zarpellon et al., 2021) | 0/100 | 21.57 | 126 | 0/100 | 126.8 | 456 | 0/100 | 929.82 | 495 | 0/100 | 11.86 | 56 | 0/100 | 131.3 | 1732 | 0/100 | 3338.97 | 16596 |
| T-BranT (Lin et al., 2022) | 0/100 | 18.62 | 142 | 0/100 | 135.22 | 1052 | 0/100 | 638.19 | 1220 | 0/100 | 9.31 | 51 | 0/100 | 113.44 | 1521 | 0/100 | 3338.47 | **7011** |
| TGAT (Seyfi et al., 2023) | 15/100 | 17.92 | **96** | 4/100 | 113.12 | **299** | 3/100 | 372.71 | 336 | 5/100 | 8.45 | 46 | 17/100 | 96.42 | 1457 | 0/100 | 1201.55 | 18442 |
| Symb4CO (Kuang et al., 2024a) | 0/100 | 19.94 | 121 | 14/100 | 109.66 | 313 | 0/100 | 497.39 | 338 | 0/100 | 10.8 | 49 | 0/100 | 134.72 | 794 | 9/100 | 1095.41 | 17837 |
| GS4CO (Kuang et al., 2024b) | 0/100 | 22.72 | 101 | 0/100 | 119.83 | 327 | 0/100 | 452.83 | 345 | 0/100 | 10.53 | 52 | 0/100 | 131.35 | 852 | 1/100 | 1137.28 | 16341 |
| DQN-BBMDP (Strang et al., 2025) | 0/100 | 21.57 | 98 | 0/100 | 117.73 | 311 | 0/100 | 448.17 | 337 | 0/100 | 10.33 | 47 | 0/100 | 128.64 | 837 | 0/100 | 1134.05 | 16259 |
| CAMBranch (Lin et al., 2024) | 9/100 | 18.11 | 97 | 2/100 | 114.53 | 317 | 0/100 | 461.83 | 359 | 2/100 | 9.07 | **41** | 0/100 | 145.09 | 1648 | 3/100 | 1143.61 | 16473 |
| MILP-Evolve (Li et al., 2025) | 2/100 | 20.58 | 103 | 0/100 | 117.65 | 327 | 5/100 | 352.5 | 324 | 2/100 | 8.79 | 48 | 0/100 | 135.7 | 1586 | 0/100 | 1163.47 | 17663 |
| *SC-MILP* (Ours) | **74/100** | **17.14** | 111 | **79/100** | **104.38** | 309 | **92/100** | **331.3** | 294 | **77/100** | **7.89** | 44 | **83/100** | **94.1** | 1385 | **87/100** | **964.79** | 14726 |

stances not solved within the time limit are treated as time-outs, and a solving time of 3600 seconds is assigned when computing average solving time statistics across instances. Code available at https://github.com/lutk3029/SC-MILP.

## 5.2. Experimental Results

**Exp-1: Performance on generated datasets.** We first compare *SC-MILP* with 15 baselines on four benchmarks; results are summarized in Table 1. Overall, *SC-MILP* consistently achieves the shortest solving time, the highest number of wins, and fewer B&B nodes in most cases.

As the primary performance metric, solving time clearly demonstrates the advantage of *SC-MILP*. Across all datasets, *SC-MILP* outperforms current SOTA MILP-Evolve by 12.36% and GCNN by 26.48% on average, with gains over MILP-Evolve of 21.25%, 20.45%, and 11.25% on easy, medium, and hard instances. Notably, most neural baselines outperform traditional FSB and RPB, confirming the efficiency of neural-based branching strategies.

Tree size reflects search efficiency and space usage. On average, *SC-MILP* explores 3.28% fewer nodes than T-BranT, the strongest baseline in terms of node count, and 26.48% fewer than MILP-Evolve. Although *SC-MILP* occasionally

generates slightly more nodes, it still achieves the lowest solving time, as overall efficiency depends on both node count and per-node computation. In particular, T-BranT's attention modules incur higher inference overhead, while our GCNN model and hybrid branching strategy could refine decisions with minimal overhead. Traditional FSB and RPB may generate fewer nodes, but remain inefficient due to expensive branching evaluations.

*SC-MILP* achieves the fastest solving time in 77.08% of instances, substantially outperforming all baselines. By difficulty, it achieves 66.00%, 75.25%, and 87.00% wins on easy, medium, and hard instances, respectively. While performance inevitably declines on harder instances, SC-MILP exhibits more robust behavior, with its performance degrading more slowly than the baselines.

Overall, *SC-MILP* simultaneously reduces solving time and tree size while achieving the most wins, validating superior efficiency and generalization.

**Exp-2: Performance on Branching Accuracy.** We evaluate top-$k$ branching accuracy ($k = 1, 3, 5, 10$) against neural baselines trained with strong branching, excluding FSB and RPB (non-learning) and T-BranT (not trained on strong branching). Results are reported in Table 2. *SC-*

*Table 2.* Imitation learning accuracy on the test sets (%).

| Model | Set Covering | | | | Combinatorial Auction | | | | Capacitated Facility Location | | | | Maximum Independent Set | | | |
|---|---|---|---|---|---|---|---|---|---|---|---|---|---|---|---|---|
| | acc@1↑ | acc@3↑ | acc@5↑ | acc@10↑ | acc@1↑ | acc@3↑ | acc@5↑ | acc@10↑ | acc@1↑ | acc@3↑ | acc@5↑ | acc@10↑ | acc@1↑ | acc@3↑ | acc@5↑ | acc@10↑ |
| Trees (Alvarez et al., 2017) | 54.7 | 74.9 | 83.7 | 93.3 | 47.7 | 69.6 | 80.1 | 91.5 | 63.4 | 90.0 | 96.7 | 99.9 | 40.6 | 53.5 | 59.0 | 65.8 |
| LMART (Burges, 2010) | 60.1 | 78.4 | 86.3 | 94.8 | 48.8 | 69.1 | 79.3 | 90.3 | 68.3 | 92.4 | 97.2 | 99.9 | 55.1 | 68.3 | 73.2 | 78.9 |
| svmrank (Khalil et al., 2016) | 59.9 | 79.1 | 86.3 | 95.0 | 58.0 | 77.6 | 86.2 | 94.0 | 68.2 | 92.0 | 97.5 | **100.0** | 55.5 | 69.3 | 74.8 | 81.6 |
| GCNN (Gasse et al., 2019) | 61.8 | 80.9 | 88.9 | 96.3 | 64.1 | 83.4 | 90.8 | 96.8 | 70.4 | 92.9 | 97.9 | **100.0** | 64.0 | 76.7 | 82.3 | 90.3 |
| FILM (Gupta et al., 2020) | 44.1 | 64.8 | 76.0 | 90.2 | 43.6 | 76.6 | 84.2 | 94.7 | 67.5 | 90.6 | 96.6 | 99.9 | 62.3 | 75.1 | 79.1 | 89.7 |
| TreeGate (Zarpellon et al., 2021) | 64.1 | 73.6 | 84.3 | 94.6 | 61.4 | 81.5 | 85.9 | 95.2 | 68.5 | 91.5 | 97.0 | 99.9 | 62.8 | 75.8 | 80.6 | 89.8 |
| TGAT (Seyfi et al., 2023) | 68.5 | 79.8 | 89.2 | 97.7 | **71.4** | 84.4 | 90.5 | 96.9 | 69.5 | 92.4 | 97.4 | **100.0** | 62.1 | 76.4 | 81.4 | 90.5 |
| Symb4CO (Kuang et al., 2024a) | 57.3 | 73.2 | 85.6 | 94.8 | 60.2 | 79.9 | 83.4 | 93.9 | 66.7 | 89.7 | 96.3 | 99.1 | 58.6 | 74.9 | 79 | 87.4 |
| GS4CO (Kuang et al., 2024b) | 56.8 | 72.7 | 87.2 | 94.5 | 61.5 | 81.4 | 85.7 | 94.3 | 67.1 | 90.9 | 95.7 | 99.4 | 59.5 | 75.6 | 80.3 | 89.2 |
| CAMBranch (Lin et al., 2024) | 60.7 | 78.6 | 87.4 | 96.2 | 63.5 | 82.7 | 90.5 | 96.8 | 68.9 | 91.2 | 96.8 | 99.9 | 63.2 | 76.4 | 81.7 | 90.6 |
| MILP-Evolve (Li et al., 2025) | 61.3 | 74.2 | 88.6 | 97.1 | 62.3 | 83.1 | 88.9 | 96.4 | 61.8 | 88.8 | 97.5 | 99.9 | 55.6 | 73.1 | 78.2 | 84.7 |
| *SC-MILP* (Ours) | **68.9** | **85.3** | **92.3** | **98.2** | 66.4 | **85.2** | **91.6** | **97.4** | **71.3** | **93.8** | **97.6** | **100.0** | **65.3** | **78.6** | **84.1** | **91.2** |

*Table 3.* Ablation study on the set covering problem.

| Model | acc@1(%) | | Easy | | Medium | | Hard | |
|---|---|---|---|---|---|---|---|---|
| | All↑ | Top 20%↑ | Time↓ | Nodes↓ | Time↓ | Nodes↓ | Time↓ | Nodes↓ |
| w/o UAMD | 64.2 | 46.7 | 6.04 | 139 | 41.16 | 1479 | 1096.43 | 34461 |
| w/o EquMD | 64.8 | 47.2 | 6.04 | 136 | 40.85 | 1474 | 1073.76 | 33897 |
| w/o PerMD | 68.1 | 51.1 | 6.00 | 119 | 37.14 | 1457 | 975.31 | 29841 |
| w/o DSCons | 65.4 | 43.3 | 6.03 | 135 | 41.33 | 1473 | 1137.06 | 33580 |
| w/o DSWeights | 66.4 | 45.7 | 6.01 | 126 | 39.82 | 1467 | 1085.46 | 31478 |
| w/o HBS | **68.9** | **51.3** | **5.95** | **107** | 37.85 | 1467 | 1011.35 | 30258 |
| *SC-MILP* (Ours) | **68.9** | **51.3** | 5.99 | 117 | **37.01** | **1452** | **953.24** | **29375** |

*Table 4.* Comparison of training on the equal-sized original dataset (ESD) versus the upstream-augmented dataset (UAD).

| Dataset | Model | acc@1 (%) | | Easy | | Medium | | Hard | |
|---|---|---|---|---|---|---|---|---|---|
| | | All ↑ | Top 20% ↑ | Time↓ | Nodes↓ | Time↓ | Nodes↓ | Time↓ | Nodes↓ |
| *ESD* | GCNN | 62.1 | 41.6 | 6.11 | 170 | 42.37 | 1479 | 1268.53 | 36417 |
| | MILP-Evolve | 61.7 | 38.7 | 10.04 | 147 | 45.88 | 1422 | 1015.83 | 30472 |
| *UAD* | GCNN | 65.4 | 43.3 | 6.07 | 149 | 41.65 | 1475 | 1154.33 | 34793 |
| | MILP-Evolve | 63.6 | 42.4 | 9.51 | 133 | 45.27 | 1419 | 982.59 | 29737 |
| | *SC-MILP* (Ours) | **68.9** | **51.3** | **5.99** | **117** | **37.01** | **1452** | **953.24** | **29375** |

*MILP* achieves the best performance on all four benchmarks. Compared with the strongest baseline TGAT, *SC-MILP* improves acc@1, acc@3, acc@5, and acc@10 by 0.4%, 5.1%, 4.0%, and 0.7% on average, respectively. Given that TGAT employs a substantially more complex model architecture, this result further underscores the effectiveness of accurately modeling upstream versus downstream semantics during training. Moreover, the high acc@5 and acc@10 indicate that strong branching variables are consistently included among top-ranked candidates, directly supporting our hybrid branching strategy.

**Exp-3: Ablation Study.** We conduct an ablation study on set covering to assess the contributions of each part of *SC-MILP*. In addition to overall metrics, we report acc@1 on the top 20% shallow nodes to evaluate upstream prediction quality. Results are shown in Table 3, including ablations that remove upstream-augmented MILP derivation (w/o UAMD), its equivalent or perturbed variants (w/o EquMD, w/o PerMD), dynamic stratified contrastive learning (w/o DSCons) or its stratified weighting (w/o DSWeights), as well as the hybrid branching strategy (w/o HBS). Further ablation results that analyze individual strategies within UAMD are provided in Appendix E.4.2.

The full model consistently achieves the best branching accuracy, shortest solving time, and smallest B&B tree in most cases, confirming the effectiveness and complementarity of all components. Removing UAMD increases solving time by 14.7% and reduces acc@1 by 4.7%, with the largest degradation caused by removing equivalent MILP derivation (w/o EquMD), while perturbed derivation (w/o PerMD) has a smaller effect. Removing DSCons leads to an 18.9%

increase in solving time and a 3.5% drop in acc@1; retaining contrastive learning without dynamic stratified weights (w/o DSWeights) partially recovers performance, highlighting the importance of stratified weighting. Removing the hybrid branching strategy (w/o HBS) does not affect intrinsic branching accuracy; its impact is mainly on efficiency, benefiting medium and hard instances.

Notably, accuracy drops more for upstream nodes: removing DSCons and UAMD decreases acc@1 by an additional 3.1% and 0.3%, confirming that our method particularly enhances upstream branching, where decisions most affect overall solving efficiency.

**Exp-4: Comparison between Upstream-Augmented and Equal-Sized Original Datasets.** To disentangle our upstream augmentation from data volume, we train all methods either on our augmented dataset or on an equal-sized original data. Results are shown in Table 4. When trained on UAD, *SC-MILP* consistently outperforms GCNN and MILP-Evolve in solving time, B&B tree size, and branching accuracy, demonstrating the effectiveness of our contrastive training strategy. For GCNN and MILP-Evolve, training on ESD yields worse performance, especially on upstream nodes, indicating that upstream augmentation is more effective than simply increasing data volume.

**Exp-5: Performance on real-world datasets.** We evaluate the performance of *SC-MILP* on real-world datasets. The results are shown in Table 5. *SC-MILP* consistently outperforms existing branching strategies, achieving an average solving-time reduction of 5.15% over MILP-Evolve, 9.86% over GCNN, and 22.83% over the original strong branching method across six benchmarks. These results demonstrate the practical effectiveness and scalability of *SC-MILP* for

*Table 5.* Performance on real-world datasets. We only report the solving time (s). Neos and Supportcase indicate MIPLIB mixed neos and supportcase, respectively.

| Model | MIK | CORLAT | Neos | Supportcase | Load Balancing | Anonymous |
|---|---|---|---|---|---|---|
| FSB | 341.53 | 147.37 | 542.01 | 374.18 | 794.68 | 472.79 |
| GCNN | 256.42 | 87.49 | 417.70 | 348.65 | 756.29 | 421.43 |
| CAMBranch | 244.59 | 82.43 | 401.85 | 321.09 | 738.76 | 403.88 |
| MILP-Evolve | 248.35 | 85.66 | 399.47 | 304.12 | 741.66 | 395.04 |
| *SC-MILP* (Ours) | **233.96** | **77.53** | **357.41** | **293.53** | **728.14** | **371.76** |

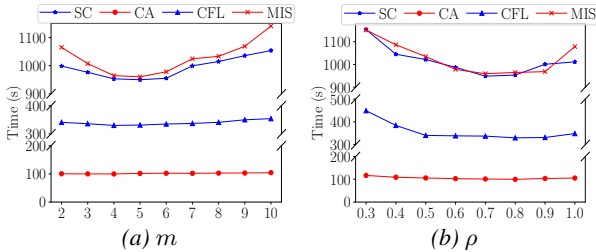

*Figure 2.* Parameter analysis. SC indicates Set Covering, CA indicates Combinatorial Auction, CFL indicates Capacitated Facility Location, and MIS indicates Maximum Independent Set.

large-scale, real-world MILP instances.

**Exp-6: Parameter Analysis.** We analyze two key hyperparameters—the group number $m$, and variable ratio threshold $\rho$—by varying $m \in [2, 10]$, and $\rho \in [0.3, 1]$ (other settings follow **Exp-1**). Results are in Fig. 2, where the analysis of contrastive temperature $\tau$ is in Appendix E.4.3. Increasing $m$ initially improves performance by better distinguishing node semantics, but performance degrades beyond $m = 4$–6 due to over-partitioning adjacent strata (Fig. 2a). Selecting $m$ in this range maximizes solving efficiency, consistent with the elbow-method results. Similarly, increasing $\rho$ applies hybrid branching to more nodes, improving accuracy but incurring higher computational cost; beyond 0.8–0.9, the marginal accuracy gain diminishes, revealing an optimal efficiency trade-off (Fig. 2b).

## 6. Conclusion

We propose *SC-MILP*, a Dynamic Stratified Contrastive Training framework for MILP branching. By grouping nodes by feature distributions and generating both equivalent and perturbed upstream MILPs, *SC-MILP* learns a discriminative model that enforces moderate separation between adjacent phases and stronger discrimination for distant ones via dynamic stratified contrastive learning. This captures fine-grained semantic differences across the B&B tree, improving branching accuracy and solving efficiency. Experiments show that *SC-MILP* achieves SOTA performance, reducing MILP solving time by 12.36% on average. Future work includes developing a neural model that generalizes across MILP types and extending *SC-MILP* to multiple solvers to broaden practical applicability.

## Acknowledgements

We thank the anonymous reviewers for their constructive comments and suggestions.This work was supported by the National Natural Science Foundation of China (Grant No. U22B2021, U24B20143, and 62572034), State Key Laboratory of Complex & Critical Software Environment (SKLCCSE), and CIE-Tianyi Cloud Research Program.

## Impact Statement

This paper presents work whose goal is to advance the field of machine learning. There are many potential societal consequences of our work, none of which we feel must be specifically highlighted here.

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

# A. Background

## A.1. Distribution Shift Between Upstream and Downstream Nodes

To examine whether the feature distributions differ between upstream and downstream regions of the search tree, we randomly sampled 1,000 nodes from each region, corresponding to depths within the shallowest and deepest 20% of the tree, respectively. All feature dimensions are normalized to zero mean and unit variance.

We assess distributional differences using the Maximum Mean Discrepancy (MMD) (Gretton et al., 2006), a kernel-based non-parametric test for measuring discrepancies between two distributions. Given upstream samples $X = \{x_i\}_{i=1}^{1000}$ and downstream samples $Y = \{y_j\}_{j=1}^{1000}$, the squared MMD statistic is computed as

$$\text{MMD}^2(X, Y) = \frac{1}{1000(1000-1)} \sum_{i \neq i'} k(x_i, x_{i'}) + \frac{1}{1000(1000-1)} \sum_{j \neq j'} k(y_j, y_{j'}) - \frac{2}{1000 * 1000} \sum_{i,j} k(x_i, y_j),$$

where

$$k(x, y) = \exp\left(-\frac{\|x - y\|_2^2}{2\sigma^2}\right)$$

is the Gaussian (RBF) kernel with bandwidth $\sigma$.

We formulate the hypotheses as:

$$H_0 : \text{The upstream and downstream samples are drawn from the same distribution,}$$

$$H_1 : \text{The upstream and downstream samples are drawn from different distributions.}$$

To determine statistical significance, we conduct a permutation test using $\text{MMD}^2(X, Y)$ as the test statistic. Under the null hypothesis $H_0$, the group labels are exchangeable. We therefore pool the upstream and downstream samples and repeatedly permute the pooled data, randomly splitting it into two sets of sizes $m$ and $n$. For each permutation, we recompute the squared MMD, yielding an empirical null distribution.

The $p$-value is estimated as the proportion of permutations whose $\text{MMD}^2$ is greater than or equal to the observed $\text{MMD}^2(X, Y)$. Using a significance level of $\alpha = 0.05$, we obtain a $p$-value of $0.003$. We thus reject the null hypothesis $H_0$ and conclude that the feature distributions of upstream and downstream nodes differ significantly.

Beyond statistical tests, Tables 3 and 4 show that the acc@1 on the first 20% of nodes differs notably from the overall acc@1, indicating the existence of distribution shift between upstream and downstream nodes and its negative effective on branch learning accuracy.

## A.2. The Cost of Obtaining Expert Samples

From our experiments, collecting 100k expert samples for the four combinatorial optimization problems (Easy level) evaluated in (Gasse et al., 2019)—namely the Set Covering Problem, Combinatorial Auction Problem, Capacitated Facility Location Problem, and Maximum Independent Set Problem—requires substantial computational time: 31.94 hours, 16.72 hours, 108.63 hours, and 70.45 hours, respectively, which is consistent with the observations in (Lin et al., 2024).

In this work, upstream-augmented MILP derivation generates approximately 30% additional training samples. Producing the same number of samples using strong branching would require around 68.3 hours, whereas our approach generates them within minutes. More importantly, samples obtained via strong branching may exhibit feature distribution shifts relative to the original data, which is undesirable for our setting. In contrast, samples generated through upstream-augmented MILP derivation preserve distributional consistency with the original samples, as they are derived from equivalent MILP formulations.

## A.3. Branch-and-Bound Framework

The *Branch-and-Bound* (B&B) algorithm is the core framework of modern MILP solvers. It maintains a search tree where:

- Each node represents a subproblem obtained from (1) by adding branching constraints.

- The LP relaxation of the subproblem provides a lower bound on its objective value.

- Nodes are *branched* if their LP solution is fractional, *pruned* if their bound is worse than the incumbent, or used to update the incumbent if feasible and better.

B&B is exact: once all nodes are processed or pruned, the best incumbent is the global optimum.

### A.4. Strong Branching

A key component of B&B is the *branching strategy*, i.e., how to choose the variable to branch on. *Strong Branching* (SB) is one of the most effective rules: for each fractional candidate variable $x_j$ with LP value $x_j^*$, SB tentatively branches by adding:

$$x_j \leq \lfloor x_j^* \rfloor \quad \text{and} \quad x_j \geq \lceil x_j^* \rceil, \tag{8}$$

and solves the LP relaxation for each branch. Let $z^*$ denote the LP bound of the current node, and $z_j^{\downarrow}$ and $z_j^{\uparrow}$ denote the LP bounds obtained from the down and up branches, respectively. The *bound improvements* are defined as:

$$\Delta_j^{\downarrow} = z_j^{\downarrow} - z^*, \quad \Delta_j^{\uparrow} = z_j^{\uparrow} - z^*, \tag{9}$$

where $\Delta_j^{\downarrow} > 0$ (or $\Delta_j^{\uparrow} > 0$) indicates that the corresponding branch yields a tighter bound, thereby potentially enabling more pruning in subsequent search. The SB score for variable $x_j$ is computed as:

$$s_j = \Delta_j^{\downarrow} + \Delta_j^{\uparrow}, \tag{10}$$

and the variable with the highest $s_j$ is selected. While SB often yields smaller search trees, it requires solving $2n$ additional LPs for $n$ candidates, making it computationally expensive.

## B. Neural Methods for MILP Solving

In recent years, the integration of machine learning techniques, particularly deep learning and graph neural networks, has attracted increasing attention in accelerating Branch-and-Bound (B&B) algorithms for mixed-integer programming. A growing body of work explores neural-based approaches to replace or enhance handcrafted heuristics at different stages of the B&B procedure. Representative studies investigate learning-based strategies for variable selection in branching (Khalil et al., 2016; Gasse et al., 2019), node selection (He et al., 2014; Labassi et al., 2022; Zhang et al., 2025), cutting plane selection (Huang et al., 2022; Balcan et al., 2022; Ling et al., 2024), variable subset selection for reoptimization (Song et al., 2020), and heuristic scheduling (Khalil et al., 2017; Chmiela et al., 2021).

In parallel, another line of research focuses on neural heuristics that directly construct high-quality feasible solutions for MILPs, rather than accelerating the B&B process itself. These methods typically generate partial assignments of decision variables to guide large neighborhood search or diving procedures. A representative example is Neural Diving (ND) (Huang et al., 2024; Liu et al., 2025; Ye et al., 2024), which learns to identify promising variables to fix in order to rapidly obtain high-quality feasible solutions.

Among these components, variable selection in branching plays a particularly critical role, as it directly determines the shape and size of the search tree and has a first-order impact on overall solving efficiency. As a result, it has been one of the most actively studied and practically impactful learning targets in neural B&B, and is the focus of this work.

Note that, following common practice and to ensure a fair comparison, we restrict our evaluation to learning-based branching methods, as other MILP strategies differ in stage, evaluation criteria, or solver configuration. Our approach specifically targets variable selection within branch-and-bound, directly influencing branching decisions. In contrast, strategies such as diving or primal heuristics operate at different stages or focus on other objectives, making their performance metrics incompatible with branching-focused evaluation. Limiting comparisons in this way ensures that the results reflect meaningful differences in the intended component of MILP solving.

# C. Theorems and Proofs

## C.1. Theorems

**Theorem C.1.** *Let the index set of integer variables in Eq. 1 be $\mathcal{I} = \{1, 2, \ldots, k\}$, and partition vectors and matrices conformably so that*

$$\mathbf{x} = (\mathbf{x}_{\mathcal{I}}^T, \mathbf{x}_{\mathcal{F}}^T)^T, \quad \mathbf{x}_{\mathcal{I}} \in \mathbb{Z}^k, \ \mathbf{x}_{\mathcal{F}} \in \mathbb{R}^{n-k}.$$

*Assume the transformation matrix*

$$T = \begin{pmatrix} B & 0 \\ F & D \end{pmatrix},$$

*where the blocks and the translation vector $\mathbf{t} = (\mathbf{t}_{\mathcal{I}}^T, \mathbf{t}_{\mathcal{F}}^T)^T$ satisfy: (a) $B \in \mathbb{R}^{k \times k}$ is a signed permutation matrix (thus each row and column has exactly one entry equal to $\pm 1$ and all others are zero. Hence $B$ is invertible and $B^{-1} = B^{\mathsf{T}}$ with integer entries); (b) $F \in \mathbb{R}^{(n-k) \times k}$ and $D \in \mathbb{R}^{(n-k) \times (n-k)}$ with $D$ invertible; (c) $\mathbf{t}_{\mathcal{I}} \in \mathbb{Z}^k$. Then we have that (1) The affine map $\phi(\mathbf{x}) = T\mathbf{x} + \mathbf{t}$ is a bijection between the feasible region of the original MILP and that of the transformed MILP, and it preserves the integrality of the integer components: if $\mathbf{x}_{\mathcal{I}} \in \mathbb{Z}^k$ then $\hat{\mathbf{x}}_{\mathcal{I}} = B\mathbf{x}_{\mathcal{I}} + \mathbf{t}_{\mathcal{I}} \in \mathbb{Z}^k$, and conversely. (2) The bijection $\phi$ induces a one-to-one correspondence between optimal solutions of the two MILPs (and of their LP relaxations).*

**Remark.** Theorem C.1 shows that the equivalence between the original and transformed MILPs holds under a broader class of linear transformations, far beyond the specific cases considered in Definition 1. In this paper, we only adopt the simpler transformation forms from Definition 1, as they are easier to implement and already provide sufficient diversity for current benchmarks.

## C.2. Proofs

### C.2.1. PROOF OF THEOREM 4.2

*Proof.* (1) *Feasible-Domain Correspondence.* Since $T$ is a diagonal matrix with entries $\pm 1$, it is invertible and satisfies $T^{-1} = T$. Thus, $\phi(\mathbf{x}) = T\mathbf{x} + \mathbf{t}$ is a bijection on $\mathbb{R}^n$ with inverse $\phi^{-1}(\hat{\mathbf{x}}) = T(\hat{\mathbf{x}} - \mathbf{t})$.

For any $\mathbf{x} \in \mathbb{R}^n$, let $\hat{\mathbf{x}} = \phi(\mathbf{x})$. Then:

$$A\mathbf{x} \leq \mathbf{b} \iff AT(\hat{\mathbf{x}} - \mathbf{t}) \leq \mathbf{b} \iff (AT)\hat{\mathbf{x}} \leq \mathbf{b} + AT\mathbf{t} \iff \hat{A}\hat{\mathbf{x}} \leq \hat{\mathbf{b}}.$$

Moreover, by the definition of $\hat{\mathbf{l}}$ and $\hat{\mathbf{u}}$ in Definition 4.1,

$$\mathbf{l} \leq \mathbf{x} \leq \mathbf{u} \iff \hat{\mathbf{l}} \leq \hat{\mathbf{x}} \leq \hat{\mathbf{u}},$$

since each component interval is correctly oriented via the $\min/\max$ construction. Hence, $\mathbf{x} \in \mathcal{P} \iff \hat{\mathbf{x}} \in \hat{\mathcal{P}}$, establishing the bijection.

(2) *Optimal-Solution Correspondence.* Let the original LP objective be $f(\mathbf{x}) = \mathbf{c}^\mathsf{T}\mathbf{x}$ and the transformed LP objective be

$$\hat{f}(\hat{\mathbf{x}}) = \hat{\mathbf{c}}^\mathsf{T}\hat{\mathbf{x}} + \gamma, \quad \text{where} \quad \hat{\mathbf{c}} = (T^{-1})^\mathsf{T}\mathbf{c}, \quad \gamma = -\mathbf{c}^\mathsf{T}T^{-1}\mathbf{t}.$$

Then for any $\mathbf{x} \in \mathcal{P}$ and $\hat{\mathbf{x}} = \phi(\mathbf{x})$,

$$\hat{f}(\hat{\mathbf{x}}) = \mathbf{c}^\mathsf{T}\mathbf{x}.$$

Suppose $\mathbf{x}^*$ is optimal for the original LP, but $\hat{\mathbf{x}}^* = \phi(\mathbf{x}^*)$ is not optimal for the transformed LP. Then there exists $\hat{\mathbf{x}} \in \hat{\mathcal{P}}$ such that $\hat{f}(\hat{\mathbf{x}}) < \hat{f}(\hat{\mathbf{x}}^*)$. Let $\mathbf{x} = \phi^{-1}(\hat{\mathbf{x}}) \in \mathcal{P}$. By the objective equality, $\mathbf{c}^\mathsf{T}\mathbf{x} < \mathbf{c}^\mathsf{T}\mathbf{x}^*$, contradicting the optimality of $\mathbf{x}^*$. The converse direction follows symmetrically. Therefore, the optimal solution sets correspond bijectively under $\phi$. $\square$

### C.2.2. PROOF OF THEOREM 4.3

*Proof.* Let $f(\mathbf{x}) = \mathbf{c}^\mathsf{T}\mathbf{x}$ and $\hat{f}(\hat{\mathbf{x}}) = \hat{\mathbf{c}}^\mathsf{T}\hat{\mathbf{x}}$ with $\hat{\mathbf{c}} = T\mathbf{c}$. As shown in Theorem 4.2, the LP relaxations are related by the bijection $\hat{\mathbf{x}} = T\mathbf{x} + \mathbf{t}$, and their optimal values satisfy $f(\mathbf{x}^*) = \hat{f}(\hat{\mathbf{x}}^*) + \text{const}$; hence the bound improvements are identical.

For strong branching on $x_j$, consider the subproblems:

$$\mathcal{P}_j^{\downarrow} = \{\mathbf{x} \in \mathcal{P} \mid x_j \leq \lfloor x_j^* \rfloor\}, \quad \mathcal{P}_j^{\uparrow} = \{\mathbf{x} \in \mathcal{P} \mid x_j \geq \lceil x_j^* \rceil\}.$$

Under $\phi(\mathbf{x}) = T\mathbf{x} + \mathbf{t}$, the image of $\mathcal{P}_j^\downarrow$ is:

$$\phi(\mathcal{P}_j^\downarrow) = \begin{cases} \hat{\mathcal{P}}_j^\downarrow & \text{if } T_{jj} = 1, \\ \hat{\mathcal{P}}_j^\uparrow & \text{if } T_{jj} = -1, \end{cases}$$

and similarly $\phi(\mathcal{P}_j^\uparrow) = \hat{\mathcal{P}}_j^\uparrow$ if $T_{jj} = 1$, else $\hat{\mathcal{P}}_j^\downarrow$.

Consequently, the optimal values satisfy:

$$z_j^\downarrow = \begin{cases} \hat{z}_j^\downarrow & \text{if } T_{jj} = 1, \\ \hat{z}_j^\uparrow & \text{if } T_{jj} = -1, \end{cases} \qquad z_j^\uparrow = \begin{cases} \hat{z}_j^\uparrow & \text{if } T_{jj} = 1, \\ \hat{z}_j^\downarrow & \text{if } T_{jj} = -1. \end{cases}$$

Thus, the set of bound improvements is preserved:

$$\{\Delta_j^\downarrow, \Delta_j^\uparrow\} = \{\hat{\Delta}_j^\downarrow, \hat{\Delta}_j^\uparrow\}.$$

Since strong branching scores (e.g., sum, product, or min of improvements) depend only on this set, the score for $x_j$ in the original problem equals that for $\hat{x}_j$ in the transformed problem. Therefore, the branching decisions correspond exactly under the mapping $\pi(j) = j$. $\qquad\square$

### C.2.3. PROOF OF THEOREM C.1

*Proof.* **(1) Bijection of feasible regions.** Since $B$ and $D$ are invertible, $T$ is invertible with

$$T^{-1} = \begin{pmatrix} B^{-1} & 0 \\ -D^{-1}FB^{-1} & D^{-1} \end{pmatrix}.$$

The affine transformation $\phi(\mathbf{x}) = T\mathbf{x} + \mathbf{t}$, therefore, has the inverse

$$\phi^{-1}(\hat{\mathbf{x}}) = T^{-1}(\hat{\mathbf{x}} - \mathbf{t}),$$

so $\phi$ is bijective on $\mathbb{R}^n$. By construction, the transformed MILP is obtained from the original one by substituting $\mathbf{x} = T^{-1}(\hat{\mathbf{x}} - \mathbf{t})$ into all constraints. Hence $\mathbf{x}$ is feasible for the original MILP if and only if $\hat{\mathbf{x}} = \phi(\mathbf{x})$ is feasible for the transformed MILP.

**(2) Preservation of integrality.** For the integer block, we have

$$\hat{\mathbf{x}}_\mathcal{I} = B\mathbf{x}_\mathcal{I} + \mathbf{t}_\mathcal{I}.$$

Because $B$ is a signed permutation matrix, $B\mathbf{z} \in \mathbb{Z}^k$ for all $\mathbf{z} \in \mathbb{Z}^k$. With $\mathbf{t}_\mathcal{I} \in \mathbb{Z}^k$, it follows that $\mathbf{x}_\mathcal{I} \in \mathbb{Z}^k \implies \hat{\mathbf{x}}_\mathcal{I} \in \mathbb{Z}^k$. Conversely, if $\hat{\mathbf{x}}_\mathcal{I} \in \mathbb{Z}^k$ then

$$\mathbf{x}_\mathcal{I} = B^{-1}(\hat{\mathbf{x}}_\mathcal{I} - \mathbf{t}_\mathcal{I}) \in \mathbb{Z}^k$$

because $B^{-1}$ is also a signed permutation matrix.

**(3) Correspondence of optimal solutions.** Let the original MILP objective be $f(\mathbf{x}) = \mathbf{c}^\mathsf{T}\mathbf{x}$. Define the transformed objective as

$$\hat{f}(\hat{\mathbf{x}}) = \hat{\mathbf{c}}^\mathsf{T}\hat{\mathbf{x}} + \gamma, \quad \hat{\mathbf{c}} = (T^{-1})^\mathsf{T}\mathbf{c}, \quad \gamma = -\mathbf{c}^\mathsf{T}T^{-1}\mathbf{t}.$$

Then for all $\mathbf{x}$ and $\hat{\mathbf{x}} = \phi(\mathbf{x})$ we have $\hat{f}(\hat{\mathbf{x}}) = f(\mathbf{x})$. If $\mathbf{x}^*$ is optimal for the original MILP, $\hat{\mathbf{x}}^* = \phi(\mathbf{x}^*)$ is feasible for the transformed MILP with the same objective value. If $\hat{\mathbf{x}}^*$ were not optimal, there would exist $\hat{\mathbf{y}}$ feasible with $\hat{f}(\hat{\mathbf{y}}) < \hat{f}(\hat{\mathbf{x}}^*)$, implying that $\mathbf{y} = \phi^{-1}(\hat{\mathbf{y}})$ is feasible for the original MILP with $f(\mathbf{y}) < f(\mathbf{x}^*)$, contradicting optimality. The reverse direction is identical. Thus, optimal solutions correspond one-to-one. $\qquad\square$

## D. Methodology Details

### D.1. Details of Stratified Node Grouping

To enable stratified node grouping, we use the node features provided by the SCIP solver, as detailed in Table 6. These features effectively characterize the solving status of each B&B node. These features capture structural and optimization-related information beyond what can be represented by node depth alone, providing a richer and more informative basis for stratified node grouping than depth-based partitioning.

All features are normalized to zero mean and unit variance, and K-means clustering is applied to partition nodes into $m$ groups, where $m$ is selected using the elbow method. This clustering is performed once on the training set, and the resulting group assignments are fixed throughout training.

Since K-means produces unordered clusters, we impose an ordering to align the strata. Specifically, for each cluster we compute a representative depth defined as the minimum depth among all nodes in the cluster. Clusters are then sorted in ascending order of this representative depth, yielding an ordered sequence from upstream (shallow) to downstream (deep) regions of the search tree.

*Table 6.* Node features for Stratified Node Grouping

| Feature Name | Type | Explanation |
| --- | --- | --- |
| depth | Integer | Depth of the node in the branch-and-bound tree (root = 0). |
| lower_bound | Float | Objective value of the LP relaxation at this node (for minimization problems). |
| num_fractional_int_vars | Integer | Number of integer variables with fractional values in the current LP solution, measuring distance to integrality. |
| max_fractionality | Float | Maximum fractional deviation among all integer variables. |
| avg_lb_shift | Float | Average shift of local lower bounds relative to global initial lower bounds across all integer variables. |
| num_conflicts | Integer | Number of conflicts triggered during inference or conflict analysis originating from this node (requires event logging). |
| branching_candidates_ratio | Float | Ratio of branching candidate variables to total integer variables, indicating potential for further branching. |

## D.2. Overview of Bipartite Graph Node Features

### D.2.1. ORIGINAL BIPARTITE GRAPH VERTEX FEATURES

Following Gasse et al. (2019), we model the MILP corresponding to each node in the Branch-and-Bound (B&B) tree with a bipartite graph, denoted by $(\mathcal{G}, C, E, V)$, where the details of these features are shown in Table 7.

*Table 7.* An overview of the features for constraints, edges, and variables in the bipartite graph $s_i = (\mathcal{G}, C, E, V)$ following Gasse et al. (2019). C = constraint vertex, E = edge, V = variable vertex.

| Type | Feature | Description |
| --- | --- | --- |
| | obj_cos_sim | Cosine similarity between constraint and objective coefficients. |
| | bias | Normalized right-hand side (RHS) value. |
| C | is_tight | Indicator of whether constraint is tight in LP solution. |
| | dualsol_val | Normalized dual value of the constraint. |
| | age | LP age since last improvement on the current vertex. |
| E | coef | Normalized coefficient $a_{ij}$ linking variable $x_j$ to constraint $c_i$. |
| | type | One-hot encoding for variable type (binary variables, integer variables, implicit integer variables, and continuous variables). |
| | coef | Normalized objective coefficient $c_j$. |
| | has_lb / _ub | Indicator whether variable has lower/upper bounds. |
| | sol_is_at_lb / _ub | Indicator whether LP solution is at lower/upper bound. |
| V | sol_frac | Fractionality of LP solution value. |
| | basis_status | One-hot simplex basis status: basic, upper, lower, zero. |
| | reduced_cost | Normalized reduced cost. |
| | age | LP age of the variable. |
| | sol_val | LP solution value of the variable. |
| | inc_val /avg_inc_val | Value/Average value in the incumbent solutions |

### D.2.2. GRAPH FEATURES OF *LT-MILP*

The relationships between MILP and *LT-MILP* vertex features are illustrated in Table 8.

Specifically, consider the linear transformation

$$\hat{\mathbf{x}} = \phi(\mathbf{x}) = T\mathbf{x} + \mathbf{t},$$

where $T$ is a diagonal matrix with entries $\pm 1$, and $\mathbf{t}_{\mathcal{I}} \in \mathbb{Z}^k$. Let the original LP relaxation of MILP (1) be

$$\min_{\mathbf{x}} \mathbf{c}^\top \mathbf{x} \quad \text{s.t.} \quad A\mathbf{x} \le \mathbf{b}, \, \mathbf{l} \le \mathbf{x} \le \mathbf{u}.$$

After the transformation, the LP becomes

$$\min_{\hat{\mathbf{x}}} \hat{\mathbf{c}}^\top \hat{\mathbf{x}} \quad \text{s.t.} \quad \hat{A}\hat{\mathbf{x}} \le \hat{\mathbf{b}}, \, \hat{\mathbf{l}} \le \hat{\mathbf{x}} \le \hat{\mathbf{u}},$$

with $\hat{\mathbf{c}} = T^\top \mathbf{c}$, $\hat{A} = AT$, $\hat{\mathbf{b}} = \mathbf{b} + AT\mathbf{t}$, and the transformed bounds are defined by $\hat{l}_j = \min\big((T\mathbf{l}+\mathbf{t})_j, (T\mathbf{u}+\mathbf{t})_j\big)$, $\quad \hat{u}_j = \max\big((T\mathbf{l}+\mathbf{t})_j, (T\mathbf{u}+\mathbf{t})_j\big)$, $\quad \forall j$.

**Dual Solution Value (`dualsol_val`).** Let $\mathbf{y} \ge 0$ denote the optimal dual variables associated with the primal constraints

*Table 8.* Relationship between MILP and *LT-MILP* vertex features. The notation $\mathbb{B} \leftrightarrow \mathbb{Z}$ denotes the potential mutual conversion between binary variables and integer variables.

| Vertex feature | MILP | *LT-MILP* |
|---|---|---|
| **Constraint vertex features (constraint $i$)** | | |
| obj_cos_sim | $C_{i,1}$ | $C_{i,1}$ |
| bias | $C_{i,2}$ | $C_{i,2} + a_i T t / \|a_i\|$ |
| is_tight | $C_{i,3}$ | $C_{i,3}$ |
| dualsol_val | $C_{i,4}$ | $C_{i,4}$ |
| age | $C_{i,5}$ | $C_{i,5}$ |
| **Edge features (edge $(i,j)$ of constraint $i$ and variable $j$)** | | |
| coef | $E_{i,j}$ | $T_{jj} E_{i,j}$ |
| **Variable vertex features (variable $j$)** | | |
| type | $V_{j,1}$ | $V_{j,1}$ or $\mathbb{B} \leftrightarrow \mathbb{Z}$ |
| coef | $V_{j,2}$ | $T_{jj} V_{j,2}$ |
| has_lb | $V_{j,3}$ | if $T_{jj} = 1$, $V_{j,3}$, else $V_{j,4}$ |
| has_ub | $V_{j,4}$ | if $T_{jj} = 1$, $V_{j,4}$, else $V_{j,3}$ |
| sol_is_at_lb | $V_{j,5}$ | if $T_{jj} = 1$, $V_{j,5}$, else $V_{j,6}$ |
| sol_is_at_ub | $V_{j,6}$ | if $T_{jj} = 1$, $V_{j,6}$, else $V_{j,5}$ |
| sol_frac | $V_{j,7}$ | if $T_{jj} = 1$, $V_{j,7}$, else 1 - $V_{j,7}$ (mod 1) |
| basis_status | $V_{j,8}$ | $V_{j,8}$ |
| reduced_cost | $V_{j,9}$ | $T_{jj} V_{j,9}$ |
| age | $V_{j,10}$ | $V_{j,10}$ |
| sol_val | $V_{j,11}$ | $T_{jj} V_{j,11} + t_j$ |
| inc_val | $V_{j,12}$ | $T_{jj} V_{j,12} + t_j$ |
| avg_inc_val | $V_{j,13}$ | $T_{jj} V_{j,13} + t_j$ |

$A\mathbf{x} \leq \mathbf{b}$. The dual of the original LP is

$$\max_{\boldsymbol{y} \geq 0} \mathbf{b}^\top \boldsymbol{y} \quad \text{s.t.} \quad A^\top \boldsymbol{y} + \mathbf{s} = \mathbf{c}, \quad \mathbf{s} \geq 0,$$

where $\mathbf{s}$ are slack variables corresponding to the (implicit) bound constraints on $\mathbf{x}$.

Under the LT-MILP transformation, each primal variable is replaced by $\hat{\mathbf{x}} = T\mathbf{x}$, where $T = \mathrm{diag}(t_1, \dots, t_n)$ with $t_i \in \{+1, -1\}$. The transformed primal problem becomes

$$\min_{\hat{\mathbf{x}}} (T\mathbf{c})^\top \hat{\mathbf{x}} \quad \text{s.t.} \quad (AT)\hat{\mathbf{x}} \leq \mathbf{b}.$$

The dual of the transformed problem is

$$\max_{\hat{\boldsymbol{y}} \geq 0} \mathbf{b}^\top \hat{\boldsymbol{y}} \quad \text{s.t.} \quad (AT)^\top \hat{\boldsymbol{y}} + \hat{\mathbf{s}} = T\mathbf{c}, \quad \hat{\mathbf{s}} \geq 0.$$

Since $T$ is diagonal with $T^{-1} = T$ and $T^\top = T$, left-multiplying the equality constraint by $T$ yields

$$A^\top \hat{\boldsymbol{y}} + T\hat{\mathbf{s}} = \mathbf{c}.$$

Comparing this with the original dual constraint $A^\top \boldsymbol{y} + \mathbf{s} = \mathbf{c}$, we see that setting

$$\hat{\boldsymbol{y}} = \boldsymbol{y}, \qquad \hat{\mathbf{s}} = T\mathbf{s}$$

satisfies the transformed dual constraint exactly:

$$A^\top \hat{\boldsymbol{y}} + T\hat{\mathbf{s}} = A^\top \boldsymbol{y} + T(T\mathbf{s}) = A^\top \boldsymbol{y} + \mathbf{s} = \mathbf{c}.$$

Furthermore:

- $\hat{\boldsymbol{y}} = \boldsymbol{y} \geq 0$ by feasibility of the original dual;

- While $\hat{\mathbf{s}} = T\mathbf{s}$ may flip signs relative to $\mathbf{s}$, its non-negativity is preserved *in the context of the transformed problem*, because the implicit variable bounds on $\hat{\mathbf{x}}$ induced by $T$ redefine the slack variables. Hence, $\hat{\mathbf{s}} \geq 0$ whenever $\mathbf{s} \geq 0$.

Importantly, this preservation relies only on the algebraic properties of $T$—specifically, that it is diagonal with entries $\pm 1$ (so that $T = T^{-1} = T^{\top}$)—and is independent of the sign pattern of $A$.

Finally, since $\hat{\boldsymbol{y}} = \boldsymbol{y}$, the dual objective value remains unchanged:

$$\mathbf{b}^{\top}\hat{\boldsymbol{y}} = \mathbf{b}^{\top}\boldsymbol{y}.$$

Therefore, under the LT-MILP transformation, both the dual solution value for each constraint and the overall dual optimum are preserved.

**Fractionality of LP Solution (`sol_frac`).** Consider an integer variable $x_j \in \mathcal{I}$ and its fractionality

$$\texttt{sol\_frac}_j = \mathrm{frac}(x_j),$$

where $\mathrm{frac}(\cdot)$ denotes the fractional part.

After the linear transformation

$$\hat{x}_j = T_{jj}x_j + t_j, \quad T_{jj} \in \{+1, -1\}, \ t_j \in \mathbb{Z},$$

the fractionality becomes

$$\texttt{sol\_\hat{}frac}_j = \mathrm{frac}(\hat{x}_j) = \mathrm{frac}(T_{jj}x_j + t_j).$$

Since adding an integer $t_j$ does not change the fractional part, we have

$$\texttt{sol\_\hat{}frac}_j = \mathrm{frac}(T_{jj}x_j).$$

Now consider the two cases for $T_{jj}$:

- $T_{jj} = +1$: $\texttt{sol\_\hat{}frac}_j = \mathrm{frac}(x_j) = \texttt{sol\_frac}_j$. The fractionality remains unchanged.

- $T_{jj} = -1$: $\texttt{sol\_\hat{}frac}_j = \mathrm{frac}(-x_j)$. Recall that

$$\mathrm{frac}(-x_j) = \begin{cases} 0, & \text{if } \mathrm{frac}(x_j) = 0, \\ 1 - \mathrm{frac}(x_j), & \text{otherwise.} \end{cases}$$

In summary, the transformed fractionality can be expressed as

$$\texttt{sol\_\hat{}frac}_j = \begin{cases} \texttt{sol\_frac}_j, & T_{jj} = +1, \\ 1 - \texttt{sol\_frac}_j \ (\mathrm{mod}\ 1), & T_{jj} = -1. \end{cases}$$

**Reduced Cost (`reduced_cost`).** Original reduced cost:

$$r_j = c_j - A_{:,j}^{\top}\boldsymbol{y}.$$

where $A_{:,j}$ is the $j$-th column of the constraint matrix, and $y$ denotes the dual vector of the constraints.

After transformation,

$$\hat{r}_j = \hat{c}_j - \hat{A}_{:,j}^{\top}\hat{\boldsymbol{y}} = T_{jj}c_j - (AT)_{:,j}^{\top}\boldsymbol{y} = T_{jj}(c_j - A_{:,j}^{\top}\boldsymbol{y}) = T_{jj}r_j.$$

Therefore, `reduced_cost` flips sign if $T_{jj} = -1$ and remains the same if $T_{jj} = 1$.

**Bounds-Related Features.** The features `has_lb` and `has_ub` are binary indicators of whether a variable has finite lower and upper bounds, respectively. Similarly, `sol_is_at_lb` (`sol_is_at_ub`) indicates whether the LP solution of a variable coincides with its lower (upper) bound.

Under the transformation $\hat{x}_j = T_{jj}x_j + t_j$, the new bounds are given by $\hat{l}_j = T_{jj}l_j + t_j$, $\hat{u}_j = T_{jj}u_j + t_j$. Since the existence of finite bounds is preserved by affine transformations, the values of `has_lb` and `has_ub` remain unchanged. However, the identity of active bounds may switch when $T_{jj} = -1$: if the solution was originally at $l_j$, after transformation it corresponds to $\hat{u}_j$, and vice versa for $u_j$.

**Solution-Related Features.** The feature `sol_val` records the LP solution value of a variable, while `inc_val` and `avg_inc_val` represent the variable's value in the current incumbent solution and the average value across all incumbent solutions, respectively.

Under the linear transformation $\hat{x}_j = T_{jj}x_j + t_j$, these values transform consistently as

$$\hat{\texttt{sol\_val}}_j = T_{jj} \cdot \texttt{sol\_val}_j + t_j, \quad \hat{\texttt{inc\_val}}_j = T_{jj} \cdot \texttt{inc\_val}_j + t_j,$$

$$\hat{\texttt{avg\_inc\_val}}_j = T_{jj} \cdot \texttt{avg\_inc\_val}_j + t_j.$$

Therefore, these features undergo the same affine transformation as the variables themselves.

### D.2.3. GRAPH FEATURES OF *RC-MILP*

After augmenting the MILP with a redundant constraint $a_r^\top x \le b_r$, a new constraint vertex is added to $\mathcal{G}$ while other vertices' features remain unchanged.

**Constraint Vertex features.** For the newly added vertex, its features such as `obj_cos_sim` and `bias` are computed from $a_r$ and $b_r$, while `is_tight` and `dualsol_val` are set to zero, since the redundant constraint does not affect the optimal solution boundary and re-solving the LP will not make it tight. `age` is also set to zero.

**Edge features.** The newly added vertex is connected to variable vertices with nonzero coefficients in $a_r$, with `coef` reflecting the normalized values.

## E. Experimental Details

### E.1. Experimental Settings

#### E.1.1. DATA GENERATION OF UPSTREAM-AUGMENTED MILP DERIVATION

**LT-MILP.** For the LT-MILP transformation, the diagonal matrix $T$ is generated by randomly selecting each diagonal entry independently from $\{-1, +1\}$. The integer translation vector $\mathbf{t}$ is constructed as follows: for variables corresponding to integer-constrained dimensions, each entry is randomly chosen from $\{-10, 10\}$; for all other entries, each is sampled uniformly from the range $[-10, 10]$. This ensures that $T$ satisfies $T^{-1} = T$ and that the transformed MILP remains equivalent to the original.

**RC-MILP.** For RC-MILP, we randomly select 10% of the original constraints (rounded to an even number) and generate new redundant constraints by summing each pair of selected constraints. These new constraints are added to the MILP to form the augmented instance.

**Perturbed MILP Formulation.** To generate perturbed MILPs, we first select 50 samples from all candidate instances for perturbed MILP derivation. Starting with a variance of 1, we progressively decrease the variance until at least 95% of the generated samples yield the same strong-branching decisions as the original instances. Finally, the variance is fixed to this value for all remaining augmented samples, ensuring that the perturbation introduces variability while preserving consistency with strong-branching choices. In our experiments, this variance was uniformly set to 0.05.

### E.2. Solver Configuration

Some baselines were originally implemented on open-source solver SCIP 6.0.1 or 7.0.1; for consistency and fair comparison, we standardize all experiments on SCIP 7.0.1 (Gamrath et al., 2020). We imposed a maximum solving time limit of 3600 seconds, allowing cut generation operations solely at the root node while disabling solver restarts. To ensure a fair comparison among the methods, we maintained all other solver parameters at their default values.

*Table 9.* The statistical description of the used datasets. In all datasets, m denotes the average number of constraints and n denotes the average number of variables.

|       | MIK | **CORLAT** | Load Balancing | Anonymous | MIPLIB mixed neos | MIPLIB mixed supportcase |
|-------|-----|--------|----------------|-----------|-------------------|--------------------------|
| $m$ | 346 | 486 | 64304 | 49603 | 5660 | 19910 |
| $n$ | 413 | 466 | 61000 | 37881 | 6958 | 19766 |

In addition, following common settings, instances that did not solve within the time limit are considered timeouts, and their solving time is set to 3600s when computing the average solving time across instances. This ensures consistency and reproducibility of the reported averages.

### E.3. The Computation of acc@k

The acc@k metric reported in Exp-2 is computed based on node-level averaging rather than instance-level or dataset-level arithmetic averaging. Therefore, the average improvement across datasets cannot be obtained by directly averaging the reported dataset-wise differences.

Specifically, each dataset contains 300 test instances. During solving, each instance generates a Branch-and-Bound (B&B) tree, where a branching prediction is made at every non-leaf node. Let $\mathcal{N}_d$ denote the set of all non-leaf nodes generated from dataset $d$, and let

$$\delta_n^{(k)} = \begin{cases} 1, & \text{if the correct branching variable is ranked within top-}k, \\ 0, & \text{otherwise,} \end{cases} \tag{11}$$

denote the acc@k indicator for node $n$.

The acc@k for a single dataset $d$ is computed as

$$\text{acc@k}(d) = \frac{1}{|\mathcal{N}_d|} \sum_{n \in \mathcal{N}_d} \delta_n^{(k)}, \tag{12}$$

i.e., the average over all non-leaf nodes rather than the average over the 300 test instances.

Similarly, the overall acc@k across multiple datasets in Experiment 2 is computed by aggregating all non-leaf nodes from all datasets:

$$\text{acc@k}_{\text{overall}} = \frac{\sum_d \sum_{n \in \mathcal{N}_d} \delta_n^{(k)}}{\sum_d |\mathcal{N}_d|}. \tag{13}$$

Therefore, the overall acc@k is not equivalent to the simple arithmetic mean of dataset-level acc@k values. This node-level aggregation ensures that each branching decision contributes equally to the final metric and appropriately accounts for differences in B&B tree sizes and computational complexity across datasets.

### E.4. Additional Experiments

#### E.4.1. COMPARISON OF DIFFERENT STRATIFIED NODE GROUPING STRATEGIES.

To evaluate the impact of different clustering strategies on branching prediction accuracy and solving efficiency, we compare our feature-driven K-means stratification with several representative alternative grouping strategies, while keeping the backbone, contrastive objective, upstream augmentation, and the number of groups $m$ fixed. Specifically, we consider: (i) depth-based grouping, using both quantile-based partitioning, which divides nodes into approximately equal-sized groups according to depth percentiles, and uniform depth intervals, which partition nodes into groups of fixed depth ranges; (ii) candidate-set-based grouping, based on the ratio of fractional variables $|C|/n$, with nodes assigned to groups according to quantiles; (iii) hierarchical clustering (Ward linkage) and Gaussian Mixture Models (GMMs), which are feature-driven strategies; and (iv) random stratification with balanced group sizes as a sanity check. The results are reported in Table 10.

Overall, grouping strategies based on unsupervised clustering methods, including our feature-driven K-means as well as GMM and hierarchical clustering, achieve comparable performance and consistently outperform strategies based on a single heuristic criterion. In contrast, uniform or random stratification yields the worst results. These observations demonstrate the

*Table 10.* Comparison of different stratified node grouping strategies.

| Model | Accuracies (%) | | | Easy | | Medium | | Hard | |
|---|---|---|---|---|---|---|---|---|---|
| | acc@1↑ | acc@3↑ | acc@5↑ | Time↓ | Nodes↓ | Time↓ | Nodes↓ | Time↓ | Nodes↓ |
| random stratification | 60.5 | 78.7 | 86.3 | 6.15 | 213 | 43.07 | 1524 | 1328.42 | 39705 |
| Depth-based (Quantile) | 65.7 | 83.1 | 90.1 | 6.03 | 147 | 41.39 | 1483 | 1137.86 | 33714 |
| Depth-based (Uniform) | 63.4 | 81.7 | 89.3 | 6.09 | 1577 | 42.04 | 1491 | 1237.76 | 36179 |
| Candidate-set-based | 67.1 | 83.9 | 90.6 | 6.02 | 145 | 38.74 | 1477 | 1074.32 | 31762 |
| Hierarchical Clustering | 68.8 | 85.3 | 92.3 | 5.98 | 116 | 37.11 | 1457 | 957.36 | 29835 |
| GMM | 68.6 | 84.9 | 92.1 | 6.00 | 117 | 37.26 | 1457 | 963.51 | 29891 |
| K-means (Ours) | **68.9** | **85.3** | **92.3** | **5.99** | **117** | **37.01** | **1452** | **953.24** | **29375** |

*Table 11.* Ablation study of upstream-augmented MILP derivation on the set covering problem.

| Weighting Scheme | acc@1(%) | | Easy | | Medium | | Hard | |
|---|---|---|---|---|---|---|---|---|
| | All↑ | Top 20%↑ | Time↓ | Nodes↓ | Time↓ | Nodes↓ | Time↓ | Nodes↓ |
| w/o UAMD | 64.2 | 46.7 | 6.04 | 139 | 41.16 | 1479 | 1096.43 | 34461 |
| w/o EquMD | 64.8 | 47.2 | 6.04 | 136 | 40.85 | 1474 | 1073.76 | 33897 |
| w/o *LT-MILP* | 66.4 | 49.2 | 6.01 | 124 | 38.66 | 1459 | 1067.31 | 30794 |
| w/o *RC-MILP* | 67.1 | 49.4 | 6.00 | 120 | 38.72 | 1468 | 1049.87 | 30479 |
| w/o PerMD | 68.1 | 51.1 | 6.00 | 119 | 37.14 | 1457 | 975.31 | 29841 |
| w/o Objective Perturbation | 68.7 | 51.0 | 6.01 | 128 | 37.21 | 1454 | 973.52 | 29716 |
| w/o Constraint Perturbation | 68.7 | 51.1 | 6.00 | 117 | 37.15 | 1458 | 968.31 | 29732 |
| w/o Dual Variable Perturbation | 68.5 | 51.1 | 6.00 | 120 | 37.06 | 1454 | 959.73 | 29514 |
| *SC-MILP* (Ours) | **68.9** | **51.3** | 5.99 | 117 | **37.01** | **1452** | **953.24** | **29375** |

effectiveness of feature-based node clustering and further indicate that our method is robust to the choice of unsupervised clustering strategy, as it consistently delivers strong performance across different clustering methods.

### E.4.2. ABLATION STUDY OF UPSTREAM-AUGMENTED MILP DERIVATION.

In this set of tests, we further conduct an ablation study on set covering to evaluate the contributions of each component in the upstream-augmented MILP derivation (UMAD). The results, presented in Table 11, include ablations that remove *LT-MILP* and *RC-MILP* from EquMD, as well as Objective, Constraint, and Dual Variable Perturbations from PerMD.

Removing all EquMD components leads to a larger degradation in both branching prediction accuracy and solving efficiency than removing either *LT-MILP* or *RC-MILP* alone, demonstrating the effectiveness of both *LT-MILP* and *RC-MILP*. Similarly, removing all PerMD components causes a performance drop than removing any single perturbation-based MILP derivation, confirming the effectiveness of Objective-, Constraint-, and Dual Variable–based perturbations.

In addition, perturbation-based MILP derivations are less effective than equivalent-based MILP derivations. However, since they are only applied during training and incur no additional inference cost, they remain valuable in practice.

### E.4.3. ANALYSIS OF PARAMETER $\tau$

We analyze the contrastive temperature $\tau$ by varying $\tau \in [0.02, 0.14]$ (other settings follow **Exp-1**). Results are in Fig. 3. The solving efficiency with varying $\tau$ exhibits a similar trend as that of $m$ (Fig. 3). Low values of $\tau$ make the model overly sensitive to hard samples, leading to unstable training, while high values weaken the contrastive separability. An intermediate $\tau$ (0.08 to 0.1) thus provides a balance between training stability and feature discriminability, yielding the best overall performance.

### E.4.4. GENERALIZATION ACROSS BACKBONE ARCHITECTURES

To further validate that the performance gains arise from the proposed *SC-MILP* training framework rather than a particular backbone architecture, we conduct additional experiments using three representative graph neural network backbones, namely GAT, GIN, and GCNN. For each backbone, we compare models trained with (*w/*) and without (*w/o*) the proposed *SC-MILP* framework.

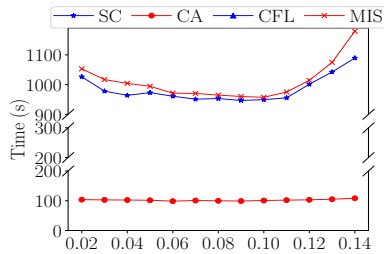

*Figure 3.* Contrastive Temperature $\tau$.

*Table 12.* Performance comparison of different backbone architectures trained with (*w/*) and without (*w/o*) the proposed *SC-MILP* framework.

| Backbone | Setting | acc@1 (%) | Easy Time | Easy Nodes | Medium Time | Medium Nodes | Hard Time | Hard Nodes |
|---|---|---|---|---|---|---|---|---|
| GAT | w/o | 64.7 | 6.53 | 249 | 48.22 | 1586 | 1264.82 | 47632 |
| | w/ | 67.9 | 6.21 | 182 | 43.58 | 1532 | 1174.70 | 38545 |
| GIN | w/o | 66.8 | 6.82 | 131 | 47.43 | 1473 | 1238.46 | 29512 |
| | w/ | 69.1 | 6.00 | 119 | 39.64 | 1431 | 1074.28 | 29278 |
| GCNN | w/o | 65.4 | 6.03 | 135 | 41.33 | 1473 | 1137.06 | 33580 |
| | w/ | 68.9 | 5.99 | 117 | 37.01 | 1452 | 953.24 | 29375 |

*Table 13.* Comparison of training time per batch and per epoch between GCNN and *SC-MILP* on the set covering problem (batch size = 128).

| Model | Training time per batch (s) | Training time per epoch (s) |
|---|---|---|
| GCNN | 0.12726 | 94.37085 |
| Ours | 0.13143 | 118.41296 |

Table 12 reports the results. Across all three backbones, incorporating *SC-MILP* consistently improves prediction accuracy while reducing solving time and Branch-and-Bound (B&B) node counts across easy, medium, and hard instances. For example, GAT achieves an acc@1 improvement from 64.7% to 67.9%, accompanied by reductions in hard-instance solving time and node count from 1264.82 to 1174.70 seconds and from 47,632 to 38,545, respectively. Similar trends are observed for GIN and GCNN.

These results demonstrate that the effectiveness of *SC-MILP* is not tied to a specific network architecture, but instead stems from the proposed training strategy, indicating strong backbone-level generalization.

## F. Efficiency Analysis of *SC-MILP*.

In this section, we analyze the additional computational cost introduced at both the training and inference stages.

**Training Overhead.** We compare the training cost of GCNN (Gasse et al., 2019) and *SC-MILP* on the set covering problem, with results shown in Table 13. Compared with GCNN, *SC-MILP* incurs only a 3.2% increase in training time per batch, indicating negligible additional computation. The training time per epoch increases by 25.5%, which is consistent with the use of approximately 30% additional upstream-augmented MILP derivations. Given the substantial gains in branching accuracy and solving efficiency, this overhead remains acceptable.

**Inference and Solving Overhead.** Our method modifies only the training stage and thus introduces no additional overhead during pure model inference; the forward inference speed is identical to GCNN. During solving, we adopt a Hybrid Branching Strategy (HBS), which applies Strong Branching to the top-$k$ candidates at upstream nodes. This incurs less than a 10% increase in per-decision cost and only at early search stages. As shown in Exp-3, the resulting improvement in upstream branching accuracy leads to a net reduction in overall solving time, fully offsetting the minor additional cost.

