# OpenReview forum: "Dynamic Stratified Contrastive Learning with Upstream Augmentation for MILP Branching"
_ICML.cc/2026/Conference — ICML 2026 spotlight_

### Official Review · Reviewer_9sjy · 2026-03-04

**Soundness:** 3
**Presentation:** 3
**Significance:** 2
**Originality:** 2
**Overall Recommendation:** 4
**Confidence:** 4

**Summary:**

The paper introduces SC-MILP, a novel neural framework designed to improve variable selection in Mixed Integer Linear Programming (MILP) solvers by addressing the semantic variation and data scarcity issues prevalent at upstream nodes of the Branch-and-Bound tree. To overcome the lack of early-stage training data without incurring the high computational costs of expert labeling, the method first groups nodes based on their feature distributions and then employs an upstream-augmented MILP derivation process that generates theoretically equivalent and perturbed instances. These augmented samples are trained using dynamic stratified contrastive learning, which learns fine-grained, depth-aware representations by pulling nodes from the same stratum together while dynamically weighting the separation of nodes from different strata based on their relative depth distance. Extensive evaluations on both synthetic and real-world benchmarks demonstrate that SC-MILP significantly outperforms existing neural branching methods by achieving higher branching accuracy, exploring fewer B&B nodes, and substantially reducing overall solving time.

**Compliance With Llm Reviewing Policy:**

Affirmed.

**Final Justification:**

The rebuttal has addressed my main concerns in their rebuttal.

**Key Questions For Authors:**

1. Recognizing that applying existing techniques to a new domain has value, what do you consider to be the most fundamentally novel algorithmic or theoretical leap in this specific combination? Are there unexpected synergies between these components that go beyond standard domain adaptation?
2. Could you provide a rationale for not applying your proposed training framework to the existing baseline models?
3. How robust is this fixed threshold when deployed on MILP instances with vastly different tree structures, candidate distributions, or anomaly patterns not seen during validation? Does the framework fail gracefully if the candidate ratio is misleading regarding the actual depth or solving stage?

**Limitations:**

Please refer to the weaknesses above.

**Strengths And Weaknesses:**

Strengths
1. Solving an important problem: The paper addresses an important bottleneck in neural combinatorial optimization: the continuous semantic variation across depths in a Branch-and-Bound tree and the data scarcity at critical upstream nodes.
2. Having some theoretical guarantees: The proposed upstream-augmented MILP derivation is backed by theoretical guarantees. Theorems 4.2 and 4.3 mathematically prove that the linear transformation-based equivalent derivations preserve both the feasible domains and the strong branching outcomes of the original MILPs.

Weaknesses
1. Limited novelty: the proposed approach lacks a fundamentally novel algorithmic breakthrough. Instead, it operates as a specialized application of existing machine learning paradigms tailored to MILP solvers. The methodology relies on standard K-means clustering for node stratification, domain-adapted data augmentation, and a variation of supervised contrastive learning to push and pull representations. While this combination of techniques is creatively applied to a real-world use case, the core reasoning is an incremental assembly of established methods rather than an innovative contribution.

2. Misaligned evaluation methodology: The authors seem to conduct a comprehensive empirical evaluation. They test their approach across four standard NP-hard MILP benchmarks (Set Covering, Combinatorial Auction, Capacitated Facility Location, and Maximum Independent Set) at varying difficulty levels, as well as on several large-scale real-world datasets. Furthermore, the study compares the proposed method against a robust suite of 15 baselines, spanning traditional branching (FSB, RPB), imitation learning, and recent reinforcement learning approaches.
However, there is a significant disconnect between the paper's primary contribution, introduced as a "Dynamic Stratified Contrastive Training Framework", and how it is evaluated. The primary experimental setup (Table 1) treats SC-MILP as a standalone model, comparing it directly against distinct architectural baselines like TGAT, T-BranT, and MILP-Evolve. A rigorous evaluation of a training framework must isolate the training strategy itself. The authors should take standard base models and demonstrate the performance delta when trained with standard supervised learning versus the SC-MILP framework. While Table 4 attempts to touch on this by training GCNN and MILP-Evolve on the Upstream-Augmented Dataset, this isolated experiment is insufficient.

3. One minor thing: The method utilizes a hybrid branching strategy at inference, relying on a hyperparameter to determine the cutoff depth between upstream and downstream nodes based on the size of the candidate set. While parameter analysis is provided, relying on a hard fraction of root candidates to dictate the branching strategy introduces a heuristic vulnerability that may not generalize perfectly across all unseen or highly anomalous MILP structures.

---

> ### Author Rebuttal · Authors · 2026-03-31
>
> We sincerely thanks for reviewing our paper and for providing valuable comments.
>
> > W1&Q1: Limited novelty: the approach largely combines existing techniques for MILP, representing an incremental integration rather than a fundamentally new algorithmic contribution.
>
> Thanks for your valueble comments. As highlighted in the fourth paragraph of our introduction, our approach is designed as an integrated framework to explicitly address the distributional inconsistencies between upstream and downstream nodes in B&B trees, rather than a disjoint combination of methods. It consists of three interdependent components:
>
> (I) stratified node grouping, which organizes nodes into meaningful stages based on their feature distributions and provides the structural foundation for subsequent modules;
>
> (II) dynamic stratified contrastive learning, which explicitly models depth-aware semantic variation by assigning contrastive weights according to group distances, thereby enhancing node representations for branching decisions;
>
> (III) upstream-augmented MILP derivation, which mitigates distributional imbalances through equivalence-based derivations with theoretical guarantees and perturbation-based derivations that ensure sufficient positive samples for contrastive learning.
>
> Through this systematic design, SC-MILP significantly improves node semantic representations and branching accuracy. Our method consistantly coutperforms all traditional branching heuristics and existing neural-based approaches across 4 synthetic benchmarks (12 settings) and 6 real-world datasets (18 in total), with particularly pronounced gains at upstream nodes.
>
> Given these advances and significant gains, we humbly believe that SC-MILP represents a clear and unique contribution over prior work.
>
>
> > W2&Q2: Misaligned evaluation: although the empirical study is extensive, it evaluates SC-MILP as a standalone model rather than isolating the proposed training framework. A proper assessment should compare standard base models trained with and without the SC-MILP framework; the limited experiment in Table 4 is insufficient.
>
> Thank you for your valuable comments.
>
> (I) Consistent with prior baselines, which couple the proposed training framework with an existing backbone to produce a standalone model for inference, SC-MILP is proposed as a training framework and instantiated with a GCNN backbone to form a complete method for evaluation. Likewise, several baselines, including FILM, TreeGate, DQN-BBMDP, CAMBranch, and MILP-Evolve, adopt GCNN as their base models, while T-BranT and TGAT are built upon GAT backbones. Therefore, evaluating SC-MILP as a standalone model and directly comparing it with prior methods is both meaningful and aligned with standard practice.
>
> (II) In addition, to directly address the reviewer’s concern, we conduct additional experiments with different backbone architectures (GCNN, GAT and GIN), comparing standard base models trained **with (w/)** and **without (w/o)** our SC-MILP framework. The results (shown in the table below) show that our framework consistently improves performance across different backbones, confirming that the gains stem from the training strategy rather than the backbone architecture.
>
> We will include these experiments in our revision to further strengthen the analysis.
>
> |Backbone|Setting|acc@1 (%)|Easy Time|Easy Nodes|Medium Time|Medium Nodes|Hard Time|Hard Nodes|
> |-|-|-|-|-|-|-|-|-|
> |GAT| w/o|64.7|6.53|249|48.22|1586|1264.82|47632|
> |GAT| w/|67.9|6.21|182|43.58|1532|1174.70|38545|
> |GIN|w/o|66.8|6.82|131|47.43|1473|1238.46|29512|
> |GIN| w/|69.1|6.00|119|39.64|1431|1074.28|29278|
> |GCNN|w/o|65.4|6.03|135|41.33|1473|1137.06| 33580|
> |GCNN|w/|68.9|5.99|117| 37.01|1452| 953.24|29375|
>
> > W3&Q3: The hybrid branching strategy at inference uses a cutoff hyperparameter (based on a fraction of root candidates), introducing a heuristic that may not generalize well to unseen or anomalous MILP instances, despite the provided parameter analysis.
>
> Thank you for your valuable comments. We acknowledge that using a fixed fraction of root candidates introduces a heuristic component. However, given the exponential growth of the search space with problem size due to the NP-hard nature of MILPs, the use of heuristics is inevitable. Such strategies are standard practice in MILP solvers; for example, many core components of SCIP, including strong branching, rely on heuristics to accelerate search and pruning.
>
> Despite this, our experiments show that the hybrid strategy is consistent effective across 4 synthetic benchmarks (12 settings) and 6 real-world datasets (18 in total), including under different parameter settings, demonstrating its consistent effectiveness and strong generalization.
> We note that the hybrid cutoff mechanism is also a practical design choice to balance efficiency and performance and is not the core contribution of our framework.
>
> We hope the above explanation alleviates your concerns.

---

> > ### Author Rebuttal · Reviewer_9sjy · 2026-04-03
> >
> > Thank you for your detailed response. I would like to raise my score to 4 now. Thanks.

---

> > > ### Author Response · Authors · 2026-04-04
> > >
> > > Thank you sincerely for your time, constructive feedback, and positive recognition of our rebuttal. We are delighted to know that your concerns have been fully resolved. We will ensure that these new results and analyses are included in the revised version. We are again truly grateful for your positive evaluation and valuable insights.

---

### Official Review · Reviewer_KM94 · 2026-03-05

**Soundness:** 4
**Presentation:** 3
**Significance:** 3
**Originality:** 3
**Overall Recommendation:** 4
**Confidence:** 4

**Summary:**

This paper proposes SC-MILP, a dynamic stratified contrastive training framework for neural branching in Mixed Integer Linear Programming (MILP) Branch-and-Bound (B&B) solving, to address three core bottlenecks of existing methods: semantic distribution shifts across B&B tree depths, severe data scarcity/imbalance at critical upstream (shallow) nodes, and the prohibitive cost of collecting high-quality strong branching samples. Extensive experiments show SC-MILP consistently outperforms 15 state-of-the-art baselines across 4 synthetic MILP benchmarks and 6 large-scale real-world datasets.

**Compliance With Llm Reviewing Policy:**

Affirmed.

**Final Justification:**

I will keep my score and have concerns regarding the limited experiments and the K-means clustering method.

**Key Questions For Authors:**

None

**Limitations:**

1. The stratified grouping is performed once on the training dataset via K-means clustering with a fixed cluster number, like m = 4 for Set Covering and m = 5 for Capacitated Facility Location. These preset steps may hinder the proposed method applied to a more dynamic environment.

**Strengths And Weaknesses:**

Strengths
1. The paper precisely identifies and addresses the most critical pain point of existing neural branching methods: the severe performance degradation at upstream B&B nodes, caused by distribution shifts across tree depths and extreme data imbalance.
2. The proposed upstream-augmented MILP derivation is a major strength. The equivalent derivation methods (LT-MILP and RC-MILP) are supported by formal theorems proving bijection of feasible regions, optimal solutions, and strong branching decisions between original and augmented instances.
3. The paper makes a novel contribution to contrastive learning for combinatorial optimization by adapting the objective to the progressive nature of B&B search. The ablation study further verifies that this stratified weighting is the core driver of performance gains, with a 18.9% increase in solving time when the contrastive component is removed.

Weakness
1. While the paper evaluates on two subsets of the MIPLIB benchmark, it does not test on the full MIPLIB 2017 dataset, which is the de facto standard for industrial MILP solver evaluation and contains highly heterogeneous instances with diverse structures.
2. The stratified grouping is performed once on the training dataset via K-means clustering, with fixed group assignments throughout training and inference. This static grouping design fails to account for the significant structural variations of B&B trees across different MILP instance types, especially for out-of-distribution instances not seen in the training set.

---

> ### Author Rebuttal · Authors · 2026-03-31
>
> We sincerely thanks for reviewing our paper and for providing valuable comments.
>
> > W1: While the paper evaluates on two subsets of the MIPLIB benchmark, it does not test on the full MIPLIB 2017 dataset, which is the de facto standard for industrial MILP solver evaluation and contains highly heterogeneous instances with diverse structures.
>
> Thank you for your valuable comments. We agree that MIPLIB 2017 is an important and widely used benchmark. However, evaluating on the full dataset remains challenging for existing neural-based branching approaches in general. Many instances in MIPLIB 2017 are extremely large-scale, which often leads to prohibitively high computational costs under standard time budgets. This difficulty is largely inherent to the nature of large-scale MILP problems: their NP-hardness, combined with diverse instance structures, makes it challenging for current neural-based piplines. Addressing such challenges typically requires specialized large-scale optimization techniques (e.g., advanced decomposition, parallelization, or problem-specific heuristics).
> Therefore, following prior studies, we adopt representative subsets for evaluation, which has become a common practice to enable controlled and meaningful comparisons.
>
> In addition, we summarize the evaluation datasets used by recent representative methods in the table below. As can be seen, most prior works focus on a limited subset of benchmarks. In comparison, our evaluation already covers a relatively broad and diverse set of datasets, providing a more comprehensive assessment of performance.
>
> | Method | Generated Dataset | MIK | CORLAT | Load Balancing | Anonymous | MIPLIB |
> | :---: | :---: | :---: | :---: | :---: | :---: | :---: |
> | GCNN | ✓ | ✗ | ✗ | ✗ | ✗ | ✗ |
> | FILM | ✓ | ✗ | ✗ | ✗ | ✗ | ✗ |
> | TreeGate | ✗ | ✗ | ✗ | ✗ | ✗ | ✓ |
> | T-BranT | ✓ | ✗ | ✓ | ✗ | ✗ | ✓ |
> | TGAT | ✓ | ✗ | ✗ | ✗ | ✗ | ✗ |
> | Symb4CO | ✓ | ✗ | ✗ | ✗ | ✗ | ✗ |
> | GS4CO | ✓ | ✗ | ✗ | ✗ | ✗ | ✗ |
> | DQN-BBMDP | ✓ | ✓ | ✗ | ✗ | ✗ | ✗ |
> | CAMBranch | ✓ | ✗ | ✗ | ✗ | ✗ | ✗ |
> | MILP-Evolve | ✓ | ✗ | ✗ | ✗ | ✗ | ✗ |
> | Ours | ✓ | ✓ | ✓ | ✓ | ✓ | ✓ |
>
> > W2&Q1: The stratified grouping is performed once on the training dataset via K-means clustering, with fixed group assignments throughout training and inference. This static grouping design fails to account for the significant structural variations of B&B trees across different MILP instance types, especially for out-of-distribution instances not seen in the training set.
>
> Thank you for your valuable comments. While the stratified grouping is performed once via K-means clustering on the training dataset, K-means is inherently adaptive, capturing feature variations in the training data and producing groups that generalize well to unseen instances.
> This is validated by our experiments in Table 10 of Appendix E.3.1, where we compare our method with several representative alternative grouping strategies, training on easy-level samples and testing on easy, medium, and hard instances. As shown in the table below, our method consistently outperforms the alternatives, with the performance gap increasing on medium and hard instances, demonstrating strong adaptability to structural variations across MILP instances, including out-of-distribution cases.
>
> Additionally, our method consistently achieves the best performance across 4 synthetic benchmarks (12 settings) and 6 real-world datasets (18 in total), demonstrating its consistent effectiveness and strong generalization across diverse problem settings.
>
> We hope the above explanation alleviates your concerns.
>
> | Model                     | acc@1 ↑ | acc@3 ↑ | acc@5 ↑ | Easy Time ↓ | Easy Nodes ↓ | Medium Time ↓ | Medium Nodes ↓ | Hard Time ↓ | Hard Nodes ↓ |
> |----------------------------|---------|---------|---------|------------|--------------|---------------|----------------|-------------|--------------|
> | random stratification      | 60.5    | 78.7    | 86.3    | 6.15       | 213          | 43.07         | 1524           | 1328.42     | 39705        |
> | Depth-based (Quantile)     | 65.7    | 83.1    | 90.1    | 6.03       | 147          | 41.39         | 1483           | 1137.86     | 33714        |
> | Depth-based (Uniform)      | 63.4    | 81.7    | 89.3    | 6.09       | 1577         | 42.04         | 1491           | 1237.76     | 36179        |
> | Candidate-set-based        | 67.1    | 83.9    | 90.6    | 6.02       | 145          | 38.74         | 1477           | 1074.32     | 31762        |
> | Hierarchical Clustering    | 68.8    | 85.3    | 92.3    | 5.98       | 116         | 37.11         | 1457           | 957.36      | 29835        |
> | GMM                        | 68.6    | 84.9    | 92.1    | 6.00       | 117          | 37.26         | 1457           | 963.51      | 29891        |
> | **K-means (Ours)**        | **68.9**| **85.3**| **92.3**| **5.99**   | **117**   | **37.01**     | **1452**   | **953.24**  | **29375**|

---

> > ### Author Rebuttal · Reviewer_KM94 · 2026-04-03
> >
> > Thank you for your response. However, I still have concerns regarding the limited experiments and the K-means clustering method. I will keep my score.

---

> > > ### Author Response · Authors · 2026-04-04
> > >
> > > Thank you sincerely for your time and constructive feedback on our rebuttal.
> > >
> > > Regarding the experimental evaluation, following existing baselines, we conduct extensive experiments on 4 synthetic benchmarks (12 settings) and 6 real-world datasets (18 in total), where our method consistently achieves the state-of-the-art performance. As shown in the comparison table in our response, our evaluation covers a broader and more diverse set of datasets and scenarios than most existing approaches.
> > >
> > > For the K-means clustering method, we have carefully compared it with multiple alternative grouping strategies (including random, depth-based, and candidate-set-based). The results consistently show that our method achieves the best or highly competitive performance, especially on medium and hard instances, demonstrating its consistent effectiveness and strong generalization across diverse problem settings.
> > >
> > > We will ensure that these additional analyses and results are included in the revised version. We are again truly grateful for your positive evaluation and valuable feedback.

---

### Official Review · Reviewer_TWgK · 2026-03-11

**Soundness:** 4
**Presentation:** 3
**Significance:** 3
**Originality:** 3
**Overall Recommendation:** 5
**Confidence:** 3

**Summary:**

The authors present an imitation learning based approach for Branch-and-Bound (B&B), "Dynamic Stratified Contrastive Learning with Upstream Augmentation for MILP Branching", which they call SC-MILP. They argue that existing such approaches (1) do not consider structural differences between upstream and downstream nodes in the branching tree sufficiently, and (2) suffer from upstream nodes being underrepresented in the training data.

The authors introduce their framework in essentially three parts:
1. Nodes are separated into so-called strata, i.e., clusters of nodes, which are then sorted by their depth from upstream to downstream. The clustering is performed by k-means using features specified in the appendix D.1.
2. Once the groups are established, upstream nodes are augmented, so the ML model can learn better to select good upstream nodes, as these "capture global optimization potential" (p. 5). Augmentation is performed by (a) generating equivalent MILP formulations to the original by applying linear transformation and by introducing redundant constraints, and (b) by applying slight perturbations (Gaussian noise) to the objective coefficients, constraint coefficients, and dual variables.
3. On the augmented stratified nodes, dynamic contrastive learning is performed. The authors use a dynamic stratified weight, i.e., different from classical supervised contrastive learning, which "pulls samples from the same group together and treats all others as negatives" (p. 5), samples from other groups are still treated as negatives, but to different degrees. This allows for a more nuanced strata separation/repulsion strength.

The authors compare their approach against a total of 15 baselines and outperform them all on average, "reducing MILP solving time by 12.36% on average" (p. 8).

**Compliance With Llm Reviewing Policy:**

Affirmed.

**Final Justification:**

The authors develop a simple idea into a comprehensive approach to alleviate the shortcomings of having upstream nodes underrepresented in imitation learning based approaches for B&B. In their framework, they combine node clustering, targeted augmentation, and dynamic contrastive learning to achieve new SOTA performance. The presentation in the main part is **sound** and **clear**, and extensive additional information is provided in the appendix.

I agree with reviewer 9sjy that this paper addresses an **important** problem. I disagree with reviewer 9sjy, however, in that the paper has limited **novelty**. I refer to the ICML reviewer guidelines, where it clearly states: "Originality need not mean wholly novel methods. It may mean a novel combination of existing methods to solve the task at hand, a novel dataset, or a new way of framing tasks or evaluating performance so as to match the needs of the user." With their "novel combination of existing methods" I see this criterion of originality/novelty clearly fulfilled.

Overall, the authors were able to resolve my concerns in their rebuttal, and I reinforce my prior recommendation to **accept** this paper.

**Key Questions For Authors:**

- Please explain why the wins in Table 1 do not sum to 100 in all columns.
- How were the percentages in the text calculated? E.g., on p. 7: "Compared with the strongest baseline TGAT, SC-MILP improves acc@1, acc@3, acc@5, and acc@10 by 0.4%, 5.1%, 4.0%, and 0.7% on average, respectively." -> these values are not immediately apparent from Table 2, as they are not simply the average of the differences between the values reported for TGAT and SC-MILP. Could you explain how they are in fact calculated?
- For Exp-4, how were the baseline methods (GCNN, MILP-Evolve) chosen? The text speaks of "all methods" (p. 8, ll. 424-425), but only results for two are reported.

**Limitations:**

None that are apparent to me.

**Strengths And Weaknesses:**

## Strengths

- The motivation for this work is clear and the authors present a simple idea that they develop into a comprehensive approach to alleviate the shortcomings of having upstream nodes underrepresented in imitation learning based approaches for B&B.
- The paper is structured well and written diligently. It includes all relevant information to understand the presented framework in the main part, including theorems and definitions. Comprehensive information is included in the appendix, including proofs, further methodological and experimental details, a discussion of neural methods for MILP solving (not for MILP branching), etc.
- The appendix also includes a statistical test to show "that the feature distributions of upstream and downstream nodes differ significantly" (p. 12), giving the motivation for this paper empirical grounding.
- By introducing perturbations in the upstream node augmentation (in objective coefficients, constraint coefficient, and dual variables, see 2.b in my summary), the model also becomes more robust (see 4.3.2)

## Weaknesses

Abstract:
- Just from reading the abstract, it was not immediately clear to me that the challenges mentioned apply specifically to learning-based methods: maybe change "these methods" to "neural methods" in line 019
- also in the abstract (first line, i.e., line 012): saying that MILP is "a fundamental NP-hard problem" is not entirely correct, as it is not a single problem, but rather a class of problems (correct in line 042 at the beginning of the introduction)

Results:
- Table 1 (p. 6): inconsistencies in the wins columns for Combinatorial Auction "Easy" and "Hard" do not sum to 100. "Easy" sums to 111 and "Hard" to 99.
- Table 2 (p. 7): for Capacitated Facility Location acc@5, GCNN actually has a higher percentage than SC-MILP (97.9 vs. 97.6) and should therefore be marked in bold as the best performing method

Minor points:
- results tables:
	- it would be nice if the methods in the results table followed the order in which they are introduced in the baselines paragraph on p. 6
	- why omit trailing zeros in the results tables? this way numbers are not aligned at the decimal point anymore
- framing/explicitness
	- p. 1-2: "However, existing neural-based branching methods largely treat all nodes uniformly, obscuring depth-dependent variations and degrading branching accuracy." -> accuracy is worse for upstream nodes, but this sounds as if accuracy degrades more downstream. Recommendation: just stick with "obscuring depth-dependent variations and branching accuracy"
	- p. 2: "invent branching algorithms" -> find a better verb (e.g., generate?), "invent" sounds like there is no basis
	- p. 7: "most neural baselines outperform traditional FSB and RPB, confirming the effectiveness of neural-based branching strategies." -> since the main metric here is solving time, "speed" or "efficiency" would be more adequate than "effectiveness"
	- p. 7: "The increasing margin on harder instances indicates stronger generalization beyond the training distribution." -> this sounds as if generalization improves with the difficulty of the instances, but performance just degrades slower than for the baseline methods, i.e., SC-MILP is more robust.
- typos
	- p. 6, l. 311, right column: nueral-based -> neural-based
	- p. 12: "its negative effective" -> "its negative effect"
- other
	- p.7: "Exp-4" is mentioned before it is introduced on p. 8 -> include the section reference to 5.2 in the brackets
	- p. 8: abbreviations ESD and UAD are introduced in the caption of Table 4, but should be introduced in the text at the first appearance
	- p. 12, A.1: first sentence is not entirely clear -> maybe a which/that is missing after "depths"? Also, not clear from the text if 1000 samples total or 1000 per quantile.

---

> ### Author Rebuttal · Authors · 2026-03-31
>
> We sincerely thanks for reviewing our paper and for providing valuable comments.
>
> > W1: Typographical and Minor Errors.
>
> We thank the reviewer for the careful reading and valuable suggestions, and we will carefully revise the manuscript to address all the points raised. In particular, we (1) correct the wins counts in Table 1 (GS4CO and DQN-BBMDP for "Easy" set to 0/100, CAMBranch for "Hard" set to 1/100); (2) replace "invent" with "devise" on Page 2; (3) revise the sentence on Page 7 to "While performance inevitably declines on harder instances, SC-MILP exhibits more robust behavior, with its performance degrading more slowly than the baselines"; (4) delete the mention of "Exp-4" in line 352 of Page 7; (5) introduce the full names of ESD and UAD in Exp-4 on Page 8, and (6) revise the first sentence of Appendix A.1 as "To examine whether the feature distributions differ between upstream and downstream regions of the search tree, we randomly sampled 1,000 nodes from each region, corresponding to depths within the shallowest and deepest 20% of the tree, respectively."
>
> We apologize for the confusion caused by these errors.
>
> > Q1: Please explain why the wins in Table 1 do not sum to 100 in all columns.
>
> Thank you for pointing out this issue. We have now corrected the wins counts in Table 1, specifically updating GS4CO and DQN-BBMDP for the "Easy" set to 0/100, and CAMBranch for the "Hard" set to 1/100 on the Combinatorial Auction benchmark. After this correction, the wins in each column sum to 100 as expected. We apologize for the confusion caused by the inconsistent "wins" in the table.
>
> > Q2: How were the percentages in the text calculated? E.g., on p. 7: "Compared with the strongest baseline TGAT, SC-MILP improves acc@1, acc@3, acc@5, and acc@10 by 0.4%, 5.1%, 4.0%, and 0.7% on average, respectively." -> these values are not immediately apparent from Table 2, as they are not simply the average of the differences between the values reported for TGAT and SC-MILP. Could you explain how they are in fact calculated?
>
> Thank you for raising this important question about the calculation methodology. We apologize for the lack of clarity in the original manuscript. Below we provide a detailed explanation.
>
> (I) For non-percentage metrics (e.g., time, nodes, wins), the relative improvement of method A over baseline B is calculated as (A−B)/B×100%. For accuracy metrics (including acc@k), the improvement is reported as the absolute difference (A−B)  since both values are already percentages.
>
> (II) However, since the reported acc@k values are derived from node-level averaging, the average improvement across datasets in Experiment 2 cannot be computed using simple arithmetic mean of the differences. Specifically, the calculation of acc@k for a single dataset is as follows: each dataset contains 300 test instances, and each instance generates a Branch-and-Bound (B&B) tree during solving. At every non-leaf node, a branching prediction is made and its acc@k is computed. The dataset-level acc@k is the average across all non-leaf nodes, not the average across 300 instances. For the overall acc@k across multiple datasets (Experiment 2), we similarly compute the average acc@k across all non-leaf nodes from all datasets, rather than taking the simple arithmetic mean of dataset-level values. This ensures each branching decision contributes equally, accounting for varying computational complexity across datasets.
>
> We are happy to incorporate these explanations in the appendix.
>
> > Q3: For Exp-4, how were the baseline methods (GCNN, MILP-Evolve) chosen? The text speaks of "all methods" (p. 8, ll. 424-425), but only results for two are reported.
>
> Thanks for your in-depth comments. In Exp-4, we selected GCNN and MILP-Evolve as representative baselines because GCNN serves as the underlying baseline model of our approach, while MILP-Evolve is the current state-of-the-art method, as demonstrated in Exp-1. We agree that the phrase "all methods" is unclear in this context; it was intended to refer only to the three methods involved in Exp-4. We will revise the sentence to "We train both GCNN and MILP-Evolve under two settings: (i) on our augmented dataset, and (ii) on an equally sized subset of the original dataset" for clarity.
>
> We hope the above explanation alleviates your concerns.

---

> > ### Author Rebuttal · Reviewer_TWgK · 2026-04-02
> >
> > Q2 follow-up:
> >
> > (I) "For accuracy metrics (including acc@k), the improvement is reported as the absolute difference (A−B) since both values are already percentages." -> Please report the absolute difference for percentage values in *percentage points*, as a percentage improvement would mean the rate of improvement, not the absolute change.
> >
> > (II) I understand your explanation (II) to be the calculation for the acc@k values, correct? What is not yet clear to me is whether the reported values in the sentence originally cited by me are to be deducible from the table or not. From the table one would get acc@1: 68.9-68.5=0.4, acc@3: 85.3-79.8=5.5, acc@5: 92.3-89.2=3.1, acc@10: 98.2-97.7=0.5, which do not correspond to the reported values 0.4, 5.1, 4.0, and 0.7. I would appreciate a short explanation in the appendix.
> >
> > ---
> > Everything else I consider resolved, thank you.

---

> > > ### Author Response · Authors · 2026-04-03
> > >
> > > Thank you for your helpful follow-up questions and for your positive feedback on our previous response. We are glad that our earlier clarifications were helpful, and we appreciate the opportunity to further clarify these remaining issues raised in your careful and thorough review.
> > >
> > > > (I) "For accuracy metrics (including acc@k), the improvement is reported as the absolute difference (A−B) since both values are already percentages." -> Please report the absolute difference for percentage values in percentage points, as a percentage improvement would mean the rate of improvement, not the absolute change.
> > >
> > > Thank you for this helpful and precise suggestion. We agree that our previous wording may have caused confusion. Following your advice, we will report absolute changes for percentage-based metrics (acc@k) explicitly in **percentage points** throughout the paper, to clearly distinguish them from relative percentage improvements.
> > >
> > > > (II) I understand your explanation (II) to be the calculation for the acc@k values, correct? What is not yet clear to me is whether the reported values in the sentence originally cited by me are to be deducible from the table or not. From the table one would get acc@1: 68.9-68.5=0.4, acc@3: 85.3-79.8=5.5, acc@5: 92.3-89.2=3.1, acc@10: 98.2-97.7=0.5, which do not correspond to the reported values 0.4, 5.1, 4.0, and 0.7. I would appreciate a short explanation in the appendix.
> > >
> > > Thank you for your careful reading and for pointing out this discrepancy. We realize that our previous explanation was not sufficiently clear. We clarify this point as follows:
> > >
> > > (a) Yes, your understanding is correct. Our explanation (II) indeed describes the procedure used to compute the acc@k values, which is consistent with existing baselines.
> > >
> > > (b) The reported values in the cited sentence can not be directly deduced from the table by computing the average acc@k across the four datasets. This is because, as explained in (II), acc@k in MILP branching is defined at the node level: each non-leaf node contributes one prediction, and acc@k is averaged over all such nodes. Different datasets (and instances) generate different numbers of non-leaf nodes, so it is not possible to compute the improvement percentage points by averaging dataset-level acc@k values. Following existing baselines, we calculate the reported improvement percentage points as the difference between two globally aggregated acc@k values, i.e., using node-level aggregation over all four datasets.
> > >
> > > We agree that this distinction was not clearly explained in the manuscript or in our previous response. Following your suggestion, we will add a concise clarification in the appendix to explicitly describe the computation procedure and explain why the reported values are not directly deducible from the table.
> > >
> > > We hope this explanation clarifies the issue and addresses your concern.

---

### Official Review · Reviewer_UtjH · 2026-03-18

**Soundness:** 3
**Presentation:** 3
**Significance:** 3
**Originality:** 3
**Overall Recommendation:** 4
**Confidence:** 4

**Summary:**

This paper studies learning-based branching for MILPs. It focuses on two issues: (1) node semantics may change across different stages of the search tree, and (2) upstream nodes are more important but more scarce in the training data. To address this, this paper proposes SC-MILP. It groups B&B nodes into strata by clustering handcrafted node features with K-means, introduces an upstream data augmentation method (including an equivalent augmentation and a perturbation augmentation), and leans node representations with a denamic stratified contrastive objective. At inference time, the method combines the learned policy with a hybrid branching strategy that uses the model more aggressively in downstream nodes while retaining stronger branching guidance for the more critical upstream part. Experiments on synthetic benchmarks and several real-world MILP datasets show improvements over several baselines in solving performance and branching accuracy.

**Compliance With Llm Reviewing Policy:**

Affirmed.

**Key Questions For Authors:**

N/A

**Limitations:**

See weaknesses.

**Strengths And Weaknesses:**

**Strengths:**

1. This paper starts from a meaningful observation that not all B&B nodes should be treated the sam, and the early branching decisions often matter more. This is a good motivation.
1. It is a natural combination of several contributions, i.e., grouping, augmentation, representation learning, and inference-time design. Especially I think the inference-time hybrid strategy does make sense.
1. The experiments are solid. The method performs well on multiple synthetic benchmarks, and the authors also make an effort to include real-world MILP datasets.

**Weaknesses:**
1. The proposed stratification method still depends on handcrafted solver features. The feature space in which clustering happens is manually designed. I do not think the paper fully justifies why these features should be regarded as an appropriate proxy for the claimed “semantic stages” of the search.
1. The paper’s motivation is based on search depth and stage, but the actual method is not stratifying by tree depth. It is stratifying by feature-induced clusters. That means there is no real guarantee that neighboring nodes, parent-child nodes, or even nodes at similar depth will be assigned to the same or adjacent strata (is there?).
1. The contrastive loss depends on $|g(i)-g(j)|$, so the cluster indices are assumed to carry an ordered notion of semantic distance. But K-means does not produce ordered clusters. The appendix resolves this by sorting clusters according to the minimum depth of nodes inside each cluster. I found this rather heuristic and not especially convincing. Using the minimum depth seems potentially unstable, especially if a cluster contains a small number of shallow outliers. Since the whole dynamic weighting mechanism depends on this ordering, I think this part needs much stronger support.
1. Some ablation studies are missing. Are the results stable under different ordering rules for the clusters. Even more directly, it would be useful to check what happens if the cluster IDs are randomly permuted before computing $|g(i)-g(j)|$.
1. LT-MILP makes sense theoretically, but RC-MILP is less convincing. The paper argues that because redundant constraints do not change the LP feasible region, strong branching outcomes should remain identical. This may not be that trivial. It would be better if the authors could provide a formal proof, or at least a more careful discussion of what kind of equivalence is actually being claimed. The perturbed MILP derivation has a similar issue.
1. The real-world experiments are still based on within-dataset train/test splits. It would be better if the authors can conduct experiments on cross-family generalization or synthetic-to-real generalization.

---

> ### Author Rebuttal · Authors · 2026-03-31
>
> We sincerely thanks for reviewing our paper and for providing valuable comments.
>
> > W1: Justification for using manually designed features for stratification.
>
> Thanks for your in-depth comments.
>
> (I) To improve clustering quality, we perform correlation-based feature selection from features provided by SCIP to construct a fixed subset (Table 7 of p.16). We further conduct a study comparing clustering performance using all features v.s. our subset (see table below), showing that the selected subset achieves better results, showing that our subsect could better capture semantic variations across the B&B tree. In addition, appendix E.3.1 shows that our feature-based clustering strategy outperforms other stratification strategies.
>
> (II) Our method consistently achieves the best performance across 4 synthetic (12 settings) and 6 real-world datasets (18 in total), demonstrating its consistent effectiveness and strong generalization across diverse problem settings.
> ||acc@1|Easy Time|Easy Nodes|Medium Time|Medium Nodes|Hard Time|Hard Nodes|
> |-|-|-|-|-|-|-|-|
> |All|65.7|6.03|133|41.43|1482|1053.87|31497|
> |Subset (ours)|68.9|5.99|117|37.01|1452|953.24|29375|
>
> > W2: Feature-based clustering may not assign structurally related nodes to the same or nearby strata.
>
> Thanks for your valuable comments. Indeed, our method intentionally uses feature-based clustering instead of depth-based stratification. As shown in Appendix E.3.1, depth-based stratification performs worse, since depth alone is an insufficient proxy for search stage: nodes at similar depths may correspond to different solver states (e.g., exploration vs. refinement), while nodes at different depths may share similar roles. We aim to group nodes with similar search contexts rather than preserve tree topology. In practice, structurally close nodes often exhibit similar features and tend to be assigned to nearby clusters. Thus, it better captures semantic search stages than depth-based stratification.
>
> > W3: Justification for using minimum-depth ordering of clusters.
>
> Thanks for your valuable comments. The ordering is not meant to recover precise stages, but to provide a coarse, monotonic proxy for contrastive weighting, roughly aligned with search progress. To address instability from shallow outliers, we conduct an ablation study by replacing minimum depth ordering with mean or maximum depth (see table below). All variants perform similarly across datasets, showing robustness to ordering choice and limited impact from shallow outliers.
>
> |Model|acc@1|Easy Time|Easy Nodes|Medium Time|Medium Nodes|Hard Time|Hard Nodes|
> |-|-|-|-|-|-|-|-|
> |Mean|68.7|6.00|118|37.15|1460|955.12|29580|
> |Max|68.9|5.99|116|37.21|1455|953.31|29420|
> |Random|66.2|6.15|135|39.72|1455|1012.53|30140|
> |Min (ours)|68.9|5.99|117|37.01|1452|953.24|29375|
>
> > W4: Effect of randomly permuting cluster IDs before computing before computing |g(i)-g(j)|.
>
> Thanks for the valuable suggestion. We conduct an ablation study by randomly permuting cluster IDs (see table above). Random permutation causes consistent performance drops in all metrics, as it breaks the monotonic alignment with search progress, making |g(i)−g(j)| no longer reflect meaningful semantic distances.
>
> > W5: More discussion for RC-MILP and perturbed MILP derivation.
>
> Thanks for this great suggestion. Due to space limitations, we provide a more detailed explanation here, but will include a full formal proof in the appendix.
> (I) RC-MILP adds a constraint that is a linear combination of existing constraints and is therefore redundant. By definition, redundant constraints do not change the feasible region ofLP relaxation, since any point satisfying the original constraints automatically satisfies the redundant one. Thus, the optimal LP value remains unchanged. Since strong branching depends on LP bound improvements after branching, the idealized strong branching scores are invariant under such redundant constraints.
> (II) For the perturbation-based derivation, the feasible region or objective may change slightly. We do not aim for exact equivalence but introduce controlled variations to diversify instances. Ablation experiments (Exp-3) confirm that both RC-MILP and the perturbations consistently improve model performance.
>
> > W6: Suggest the authors conduct experiments on cross-family or synthetic-to-real generalization.
>
> Thanks for this insightful suggestion. We agree that cross-family and synthetic-to-real generalization are important for evaluating the robustness of neural branching methods. As noted in Footnote 3 (P. 19), our experimental protocol follows Turner et al. 2023 to ensure fair comparison with prior work, which adopts within-dataset splits. We would like to clarify that cross-family generalization is inherently challenging for existing neural branching methods in general. Nevertheless, we agree it is an important direction and will explore more generalizable representations and training strategies in future work.

---

> > ### Author Rebuttal · Reviewer_UtjH · 2026-04-04
> >
> > Thank you for your response. I would like to maintain my positive rating.

---

> > > ### Author Response · Authors · 2026-04-04
> > >
> > > Thank you sincerely for your time, constructive feedback, and positive recognition of our rebuttal. We are delighted to know that your concerns have been fully resolved. We will ensure that these new results and analyses are included in the revised version. We are again truly grateful for your positive evaluation and valuable insights.

---

### Decision · Program_Chairs · 2026-04-30

**Decision:**

Accept (spotlight)

**Comment:**

Reasons to accept this paper:

1. The main idea of the work involves finding better ways to train models for branching in a MILP. The two main ideas are both novel and interesting involving stratified contrastive learning, and the augmentation of "missing" data. These are ideas that could have an impact beyond solving MILPs, so I think it is a nice paper to accept.

2. The experimental results are rather strong, showing good performance on a standard benchmark. It might have been nice to have more here, but it is sufficient.

3. The presentation of the paper is good and the method is clear.

The authors do not indicate whether they will make their code available to others. I hope they will.